# Two-photon imaging of soliton dynamics

**Łukasz A. Sterczewski** [1] ✉ **& Jarosław Sotor** [1]

Optical solitary waves (solitons) that interact in a nonlinear system can bind and form a structure similar to a molecule. The rich dynamics of this process have created a demand for rapid spectral characterization to deepen the understanding of soliton physics with many practical implications. Here, we demonstrate stroboscopic, two-photon imaging of soliton molecules (SM) with completely unsynchronized lasers, where the wavelength and bandwidth constraints are considerably eased compared to conventional imaging techniques. Two-photon detection enables the probe and tested oscillator to operate at completely different wavelengths, which permits mature near-infrared laser technology to be leveraged for rapid SM studies of emerging long-wavelength laser sources. As a demonstration, using a 1550 nm probe laser we image the behavior of soliton singlets across the 1800–2100 nm range, and capture the rich dynamics of evolving multiatomic SM. This technique may prove to be an essential, easy-to-implement diagnostic tool for detecting the presence of loosely-bound SM, which often remain unnoticed due to instrumental resolution or bandwidth limitations.

Since the discovery in hydrodynamic systems[1], solitary waves also known as solitons have expanded into diverse areas of science due to the unique way they analogize matter and waves. Arguably, a significant fraction of soliton research has been centered around nonlinear optical systems such as fibers[2] or microvavity resonators[3,4]. This is because optical solitons do not spread out during propagation and exhibit robustness against perturbations; therefore they frame the core concept in optical pulse generation. Solitons also have the striking ability to form a stable bond between pairs or groups referred to as SMs[5,6]. The binding force[7,8] depends on the atomic spacing $\tau$ (temporal separation, TS), while the intramolecular phase $\Delta\varphi$ mostly governs the attraction/repulsion effect. Even more sophisticated physical systems like molecular complexes[9] or molecular crystals[10] can form via SM collisions. Despite advances in mathematical modeling[11], the understanding of these complex inter-soliton interactions still appears to be in infancy.

The rich landscape of nonlinear dynamics and numerous analogies to condensed-matter physics fuel intensive research in this field with much focus on studying the evolution dynamics and transient behavior of SM formation[12,13]. It stems from the SM application potential to optical memories, buffers[14], or telecommunication to surpass the limitation of classical binary coding schemes[15,16]. In such scenarios, one would like to tailor the SM spacing and phase on demand[17–20] rather than rely on its uncontrolled organization, which necessitates a thorough characterization of transient short-lived SM states.

SM characterization techniques differ significantly in obtainable scan rates. Time-averaged (second-scale, Hz-rate) SM studies are performed with a first-order intensity autocorrelator (IAC) along with an optical spectrum analyzer (OSA)[21], which are suitable mostly for steady-state phenomena like stable, tightly-bound SM. The highest scan rates (shot-to-shot, sub-GHz) are offered by the celebrated Dispersive Fourier Transformation (DFT) technique[22,23], which temporally stretches a laser pulse in a linear dispersive medium (such as optical fiber) to map the time domain to the frequency domain on a pulse-by-pulse basis[12,13,18,24,25]. Temporal imaging with a time lens[26] has also gained an established position as a tool for single-shot laser pulse diagnostics[27,28]. Unfortunately, beyond the 1–2 μm window, fiber losses or availability of suitable phase modulators (for the time lens) may render both techniques impractical. Also the spectral resolution of the two techniques (typically ~GHz) may be insufficient to probe loosely-bound SMs (TS on the order of ns). A much simpler variant of both methods – direct analysis of laser pulses on an oscilloscope without temporal or dispersive stretching – completely ignores the molecular phase and fails to resolve solitons spaced by ps due to electrical bandwidth limitations.

[1]Faculty of Electronics, Photonics and Microsystems, Wroclaw University of Science and Technology, Wyb. Wyspianskiego 27, 50-370 Wroclaw, Poland.
✉e-mail: lukasz.sterczewski@pwr.edu.pl

To fill a niche between the two timescales, recent works have proposed to adapt the phase-sensitive electric field cross-correlation (EFXC) technique[29] often referred to as coherent optical sampling[30] or dual-comb spectroscopy (DCS)[31] for imaging the dynamics of solitons in microresonators[3,4]. However, high scan rates with EFXC are obtainable only with multi-GHz repetition-rate sources[3,4]. In the case of fiber laser cavities, which constitute a majority of SM generators, the aliasing (Nyquist) limitation strongly constrains the observable optical bandwidth due to sources' low, MHz repetition rates, and restricts frame rates to the 10's–100's of Hz range. The need for phase locking between the lasers also adds a layer of complexity. The greatest difficulty, however, results from the need of a second, spectrally-matched laser, which may be impractical to implement at exotic wavelengths.

In this work, to bypass these limitations and unlock the kHz- to sub-MHz rate imaging potential, we adapt the non-interferometric intensity cross-correlation (IXC) technique[32–35] to the problem of dynamic soliton imaging. Two-photon dual-comb IXC (or simply IXC throughout the rest of the text) probes SM dynamics with eased restrictions on the laser design, operation wavelength and stability. Instead of nonlinear crystals[36,37] that require optical phase matching, the imaging technique builds on the two-photon detection ranging concept by Wright et al.[38]. Unlike EXFC, the IXC technique lifts the requirements of spectral overlap and phase lock between the two lasers: a pair of free-running sources with different wavelengths and offset repetition rates can be used instead. Only their photon energies must add to satisfy the 2-photon-absorption (2PA) detection criterion, which grants access to probing lasers in emerging spectral regions. Such an implementation of IXC may also extend the wavelength capabilities of techniques that retrieve the pulse temporal intensity profile (not just IAC) like FROG[39] or SPIDER[40], or speed of wavelength-agile cross-correlation FROG (XFROG)[41]. If that of the probe pulse in IXC is characterized and known, the tested pulse profile one can be retrieved via deconvolution[32].

## Results

### Two-photon intensity cross-correlation

To image the SM behavior in an exemplary laser-under-test (LUT) cavity, we have probed it by a second mode-locked probe laser (referred to as a local oscillator, LO). On average, the lasers had tens of mW of optical powers and offered sub-ps pulse widths. The experimental setup is shown in Fig. 1a. Both sources were combined with a fiber coupler to jointly illuminate a commercial bias-free Si photodiode (PD, Thorlabs FDS02), which exhibits a 2PA response above ~1100 nm. To prevent one-photon absorption from occurring (i. e. due to residual above-bandgap pump or spurious light generated via nonlinear frequency conversion), we have incorporated a fiber long-pass filter (LPF) before the photodetector. The LUT repetition rate was $f_{r,1} \approx 100.04$ MHz, while the probe laser with $f_{r,2} = f_{r,1} + \Delta f_r$ was equipped with a tunable delay line to vary the cavity length and hence adjust the repetition rate difference $\Delta f_r$ governing the IXC scan rate. The basic principle of the the IXC technique (Fig. 1b) resembles conventional DCS (EFXC), where a time lag of $\Delta T \approx \Delta f_r / f_r^2$ that linearly increases from pulse pair to pulse pair[42] stroboscopically samples the waveform over optical delays between 0 and $1/f_r$. In the case of DCS, measured is the phase-sensitive $E$-field cross-correlation also known as the interferogram, which relates the effective (ps/fs), and laboratory time ($\mu$s/ns) time scales by the temporal magnification factor (TMF) $m = f_r/\Delta f_r$. However, the Nyquist frequency located at $f_r/2$ sets a limit on the maximum probed optical bandwidth $\Delta\nu$ expressed as $\Delta f_r \leq f_r^2/2\Delta\nu$, above which the interferogram is badly distorted. Here, the full − 20 dB bandwidth $\Delta\nu = 7.5$ THz (60 nm) can be probed with EFXC at rates $\Delta f_r \leq 670$ Hz. In IXC, the sampling process remains the same, however, the interaction relies on multiplying the combs' $E$-field intensities[32] $I_{1,2}(t) = |E_{1,2}(t)|^2$. The measured quantity is simply the intensity cross-correlation (⋆): $\langle I_1(t)I_2(t+\tau)\rangle = \int_{-\infty}^{+\infty} I_1(t)I_2 (t+\tau)\, \mathrm{d}t = I_1(t) \star I_2(t)$, where $\tau$ represents the lag scanned by the asynchronous interaction. As will be shown later, the bandwidth limitations of IXC are not as strict as in EFXC, and can be exceeded multiple times the conventional limit (see "Methods").

Background-free 2PA in the semiconductor photodetector (Fig. 1c) produces a squared-intensity electrical response when a pair of photons with energies between half the bandgap and the bandgap reaches its surface (ideally tightly focused). The low efficiency of this $\chi^{(3)}$ process implies that only high peak powers (when two pulses temporally overlap) are detected. A major advantage is that the 2PA detection process does not require a polarization or wavelength match. Therefore, as long as the comb's photon energies fall into a suitable range, an IXC signal can be produced. Once converted into the

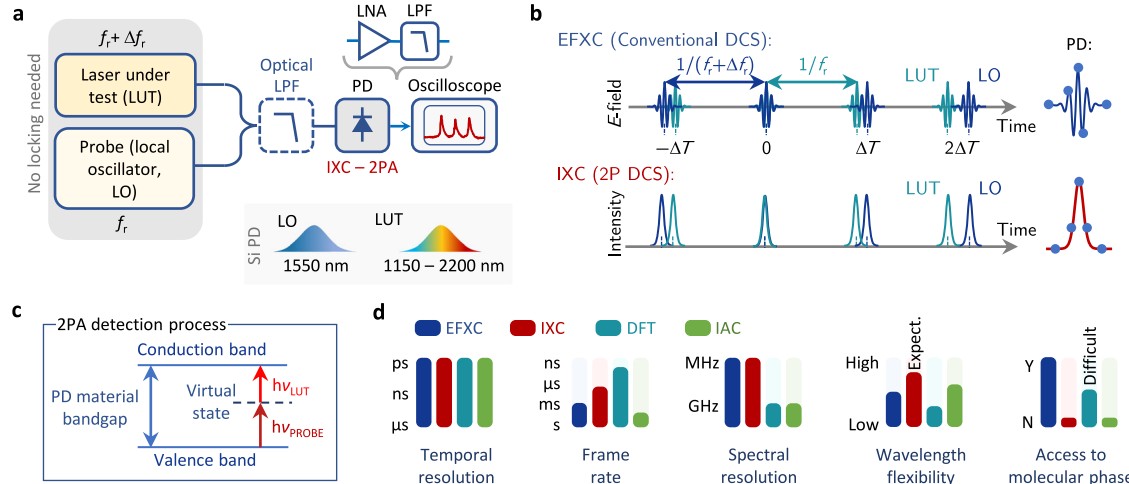

**Fig. 1 | Principles and experimental realization. a** Conceptual schematic showing the laser under test (LUT) combined with a probe. After the long-pass-filtered (LPF) optical signal is measured on a photodetector (PD), a low-noise amplifier (LNA) followed by a low-pass filter (LPF) are used for signal conditioning prior to sampling by an oscilloscope. **b** Comparison of sampling in the EFXC, and IXC. The lag increases linearly from pulse-pair to pulse-air to produce EFXC and IXC signals. **c** Energy diagram of the 2PA detection process. **d** Chart-style comparison of existing SM characterization techniques in conventional realizations with the IXC technique for a MHz-repetition-rate pulsed laser. When comparing the frame rate in dimensionless units irrespective of the laser repetition rate, it is better to use a figure-of-merit (FOM) criterion here defined as FOM = $2\Delta\nu\Delta f_r/f_r^2$ (in units of Hz²/ Hz²). For a Nyquist-limited sampling technique, is it lower or equal to 1, while for IXC it can reach values of 100 or more. Non-sampling techniques like DFT or IAC have a theoretical frame rate of $\Delta f_r = f_r$, which yields a FOM = $2\Delta\nu/f_r2$ approaching infinity for large optical bandwidths $\Delta\nu$.

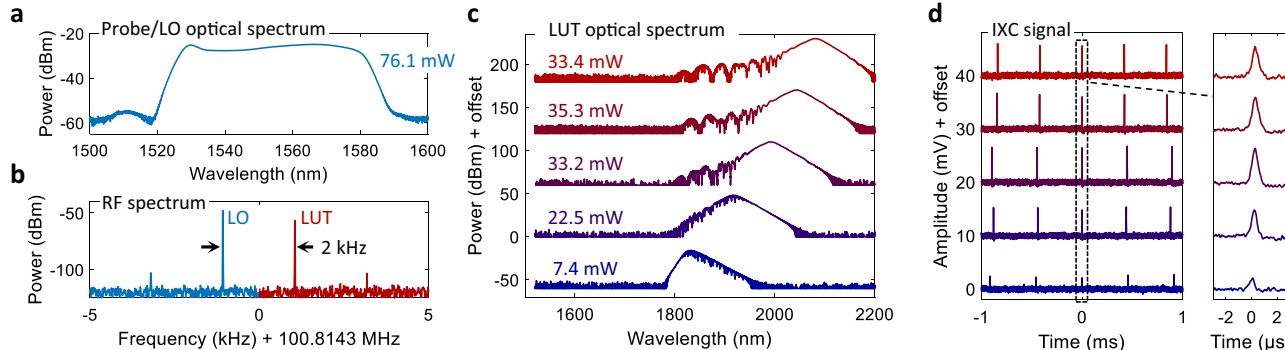

**Fig. 2 | Intensity cross-correlation between combs operating in different spectral regions. a** Optical spectrum of the probe/LO laser. After amplification, the optical power is 76.1 mW. **b** Radio-frequency beat notes of the two combs detected by the Si detector when the LUT operates at 2100 nm. **c** Tunable optical spectra of the LUT due to the soliton self-frequency shift (SSF) process. **d** IXC signal measured with an oscilloscope for LUT spectra corresponding to those in panel (**c**).

electrical domain, the signal should be conditioned by a low-noise amplifier (LNA), and a low-pass-filter (LPF) with $f_r/2$ cut-off to isolate the cross-correlation signal from the laser pulse trains prior to sampling by an oscilloscope. An example recording in given in Supplementary Video 1. For a more detailed mathematical description of the IXC technique, please see Methods.

## Weak wavelength constraints of IXC

The unique feature of IXC is its wavelength agility. To experimentally prove it, we employed a 1550 nm laser (Fig. 2a) to probe another longer-wavelength oscillator emitting pulses in the 1800–2100 nm range (Fig. 2b, c). For clarity of the demonstration, both sources were operated in the soliton singlet state, which yielded a clean IXC peak when the pulses coincided (Fig. 2d). This repeats periodically every $1/\Delta f_r$ seconds here equal to $(2\,\mathrm{kHz})^{-1} = 500\,\mathrm{ms}$, and produces a profile corresponding to the intensity cross-correlation between the pulses (see Methods for mathematical details). We would like to emphasize here the applicability of the technique to other spectral regions beyond the near-infrared. Provided a suitable two-photon detector (i.e. InGaAs or other semiconductor structures), it will be possible to employ technologically-mature thulium or holmium lasers and amplifiers to probe weaker mid-infrared oscillators. This is because the high-power probe/LO laser can provide detection gain for weaker mW-class sources to produce a sufficiently strong IXC signal.

## Scan rates and temporal resolution

With IXC, imaging SM dynamics over large optical bandwidths at high frame rates is possible beyond the aliasing limit of dual-comb interferometry, albeit at the expense of lowered temporal resolution. This counter-intuitive ability results from the practical condition that slightly chirped (non-transform-limited) pulses experience enhanced temporal overlap which broadens IXC features[38]. Since they occupy a smaller RF bandwidth, the aliasing constraints are relaxed (see "Methods" for more details). This feature enables us to directly illuminate the photodetector with broadband lasers without the need for narrow optical band-pass filtering, which complicates conventional EFXC.

Since the ultimate goal is almost always high imaging speed to fully capture the SM evolution trajectory, when comparing different SM diagnostic techniques one should consider the number of cavity round-trips required to produce a single frame rather than absolute time units. For instance, MHz scan rates obtained in imaging GHz-level $f_r$ microcavities[3,4] require $10^3$ round-trips per frame. In relative terms, imaging $f_r = 100\,\mathrm{MHz}$ fiber laser cavities with $\Delta f_r = 100\,\mathrm{kHz}$ frame rates is equivalent. This is what can be obtained using the IXC technique even for low $f_r$ cavities, as shown in Fig. 3a. We need to underline,

however, that if the soliton evolution rate exceeds the frame rate, one has to resort to single-shot imaging techniques like DTF[12], time-lens[26] or even a combination of both[28] to capture all details of the soliton evolution trajectory.

In this context, worth studying is also a single-frame IXC temporal resolution limit, which governs the ability to identify tightly-bound SM atoms and sets another limit on $\Delta f_r$. The requirement to low-pass filter the electrical IXC signal at $f_r/2$ implies that only features spaced in laboratory time by $\delta t_{lab} > 2/f_r$ can be resolved, which given the TMF, yields a temporal resolution limit $\delta t = 2\Delta T = 2\Delta f_r/f_r^2$. For a $f_r = 100\,\mathrm{MHz}$ laser, this corresponds to $\delta t = 2\,\mathrm{ps}$ for $\Delta f_r = 10\,\mathrm{kHz}$. It is therefore clear that the IXC technique is favored for probing larger temporal separations with high frame rates (where DFT struggles due to bandwidth and resolution limitations), while tightly-bound SMs entail scanning at lower rates. An experimental verification of these derivations is shown in Fig. 3b. A triatomic SM with a 6.45 ps TS has clearly resolvable atoms even when exceeding the aliasing limit ~ 20.8 times. In the laboratory time scale, the atoms are spaced by ~50 ns, which is $2.5 \times$ the resolution limit ($\delta t = 2.58\,\mathrm{ps}$). Obviously, distinguishing features in the near-resolution-limit regime may be difficult, because the significant fall time of the IXC signal due to the LPF lowers the peak contrast. It should be also noted that the ultimate resolution limit of the technique, considering the LO pulse width $\sigma_{LO}$, and its jitter $\sigma_J$, should obey the root mean square sum law, i.e. the obtainable resolution will be $\sigma_t = \sqrt{\delta t^2 + \sigma_{LO}^2 + \sigma_J^2}$.

## IXC for laser diagnostics

Notably, IXC can serve as a practical laboratory diagnostic tool to detect the presence of multiple pulses in a laser cavity. The $0-f_r$ delay scan range with a corresponding $f_r$ resolution limit analogous to a mechanical interferometer displacement on the order of meters for fiber laser cavities, is far beyond most DFT implementations, high-resolution OSAs, and even long-range autocorrelators. In many biomedical imaging applications like two-photon fluorescence[43], pulse parameters such as peak power are derived solely from short-range (<50 ps) intensity autocorrelation traces. If loosely-bound SMs form inside the cavity (and hence there is more than one pulse per round-trip), these assumptions may be completely invalid. IXC diagnostics addresses these challenges, as shown in Fig. 3c. Interestingly, probed is not a simple di- or tetratomic SM, but a recently discovered pair of bound pulses referred to as a 2+2 soliton molecular complex (SMC)[9] with distinct intra- and inter-molecular bonds. The intra-molecular TS is 854 ps, while the inter-molecular TS is 88 ps, which are in opposite ratios compared to those observed in ref. 9. Resolving such widely-spaced pulses would practically require a sub-GHz resolution, which is rarely possible with typical OSAs or IAC.

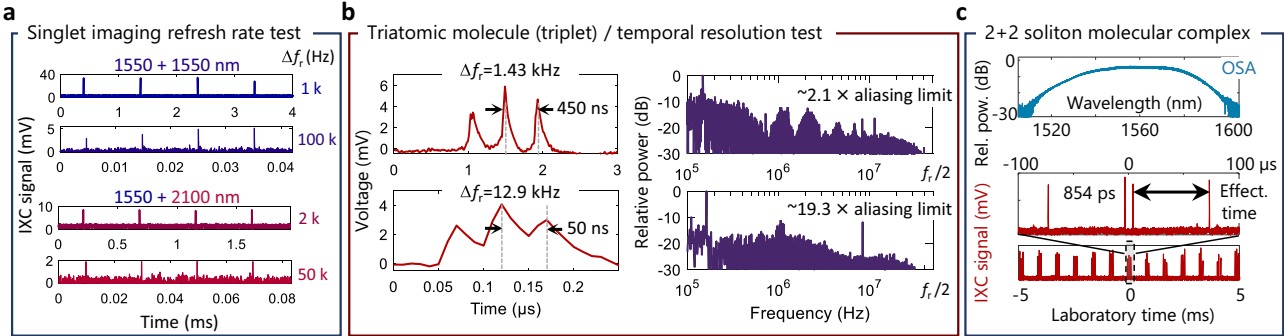

**Fig. 3 | Scan rate and temporal resolution test. a** Agility in obtainable scan rates for lasers operating at the same, and different wavelengths. The lowered signal level at higher rates results from the lower number of coincidence/overlap events when a single IXC trace is produced. **b** Temporal resolution test that shows operation above the aliasing limit. A soliton triplet (triatomic molecule) is imaged. Even exceeding the aliasing limit almost 20 times still yields resolved pulses. Shown also is the power spectrum of the IXC signal, which is mostly concentrated in the low-frequency region rather than close to the Nyquist frequency $f_r/2$. **c** 2+2 soliton molecular complex (SMC) probed by the IXC technique. The soliton pairs are separated by 88 ps, while the atom separation is 0.854 ns, which practically requires sub-GHz spectral resolutions to detect its presence. Such features are not resolved by a 50 pm (~6 GHz) resolution optical spectrum analyzer (OSA). Abbreviation: Rel. pow. − relative power.

## Two-photon imaging of SM dynamics

To demonstrate the extended-time IXC imaging capabilities, we visualize the evolution of SMs produced by a mode-locked fiber laser with $f_r = 100.04$ MHz repetition rate covering $\Delta v > 60$ nm of −20 dB optical bandwidth centered at ~1550 nm. High pump levels (>500 mW) lead to an excitement of tightly-bound tri- and tetratomic SMs, of which only the first was indefinitely stable. The fine intramolecular motion of these SM states was probed by IXC with 270 fs, and 230 fs temporal resolution (defined by $f_r/2$) in the soliton triplet and quadruplet case, respectively. Figure 4a shows the IXC signal (intensity cross-correlogram) sampled by a 12-bit oscilloscope. Subsequent zoomed panels reveal a sequence repeating with $\Delta f_r = 1.35$ kHz, which appears as three isolated pulses spaced by 500 ns. In this experiment, the probe laser (Fig. 2a) was completely unsynchronized with the LUT, as Hz-level $\Delta f_r$ stability on a second timescale yielded a relative timing uncertainty in the $10^{-3}$ range. A waterfall-type sequence of frames sampled $\Delta f_r$ per second yielded an IXC image, where the vertical ($y$-axis) was scaled according to the TMF, analogous to applying a co-moving time frame[3,12].

In the triatomic molecule case (Fig. 4b), one can observe weak intermolecular motions from the nominal 6 ps spacing. Nevertheless, the trajectories drawn by the atoms are clear and saturated, as zoomed in Fig. 4c. This picture finds confirmation in the optical spectrum measured with an OSA (Fig. 4d), which displays a regular modulation pattern with a 1.2–1.3 nm (~161 GHz) period. The simultaneously measured radio frequency (RF) spectrum (Fig. 4e) does not show any anomalies.

In contrast, the unstable tetratomic molecule (Fig. 4f) exhibits rich nonlinear dynamics reminiscent of mode competition in semiconductor lasers. Almost never do all four pulses have comparable intensities. Instead, whenever some pulses become stronger, others weaken, as plotted in zoomed-panel Fig. 4g. Notably, the TS of ~2.3 ps barely fluctuates except for minor drift of the 4-th atom above $60 \cdot 10^6$ round-trips. This noisy behavior finds confirmation in the optical spectrum (Fig. 4h), which lacks a regular modulation pattern and cannot be used to explain the laser state by itself. Throughout the duration of the scan, the soliton quadruplet was rapidly evolving, which corrupted the fringe contrast in the optical spectrum. Example four consecutive IXC frames are given in Fig. 4i.

## Microsecond-time imaging of soliton-rain-like phenomena

To demonstrate the rapid temporal dynamics capabilities of IXC imaging and agility in obtainable frame rate, we have probed a soliton-rain-like event[44] with $\Delta f_r = 83$ kHz yielding a horizontal resolution of ~1200 cavity round-trips (12 μs). Increased laser cavity losses (due to fiber bending) accompanied by high pump powers give rise to a significant quasi- continuous-wave (CW) component in the optical spectrum, which in turn mediates long-range soliton interactions. This laser state corresponds to a weakly mode-locked regime when low-intensity light filtering is not efficient to prevent multiple unlocked pulses from existing. Four IXC frames shown in Fig. 5a display a quasi-repeating yet evolving pattern with characteristic intensive regions (i.e. around 40 μs) referred to as the condensed phase surrounded by a soliton flow part. The soliton rain process resembles rain droplet formation from a vapor cloud, where spontaneously created rain droplets move toward the condensed phase like they were falling into the sea to next evaporate and start the process again[44]. Magnification of one of the frames reveals that multiple pulses spaced by approximately 90 ns exist during a round-trip. Like before, even ~124 × above the aliasing limit, the IXC signal, and its frequency spectrum (Fig. 5b) look clean and undistorted. The quasi-harmonic IXC spectrum has a regular structure with broad peaks, which reflect the soliton position timing jitter. The peak locations are multiples of ~11 MHz related to the ~90 ns laboratory time soliton spacing (75 ps effective). Except for the aforementioned CW component, the optical spectrum (Fig. 5c) does not have any special features. The RF spectrum (Fig. 5d), in contrast, is strongly modulated with multiple sidebands, which is a signature of the rich soliton rain dynamics at extended time scales. It is clear that IXC directly images the temporal SM dynamics, as opposed to providing time-averaged measurements obtained using swept optical- or RF spectrum analyzers.

## Optical power requirements

Although the Si photodiode requires watts of peak optical power to produce a detectable signal (sub-nJ pulse energy for sub-ps durations), much higher sensitivity two-photon detectors can be used instead. For instance, quantum-well laser structures have proven to exhibit 2PA sensitivities orders of magnitude higher than semiconductor photodiodes[45]. For the 1550 nm range, excellent performance is offered by multi-quantum-well Fabry-Pérot laser diodes with nominal emission wavelengths at 1.3 μm as shown in Fig. 6 (see "Methods" for device details). Whereas for the Si photodiode the sensitivity defined as a product of the peak LUT power and average LO power[45] amounts to ~ $4.4 \cdot 10^6$ mW², using an unbiased laser diode as a detector yields an improvement to $2.4 \cdot 10^3$ mW². In other words, we can probe μW average power level pulses instead of mW with corresponding fJ pulse energies. This is shown in Fig. 6a, where 9 μW, 0.5 ps-long pulses are probed by a 15 mW LO. This major sensitivity improvement offered by

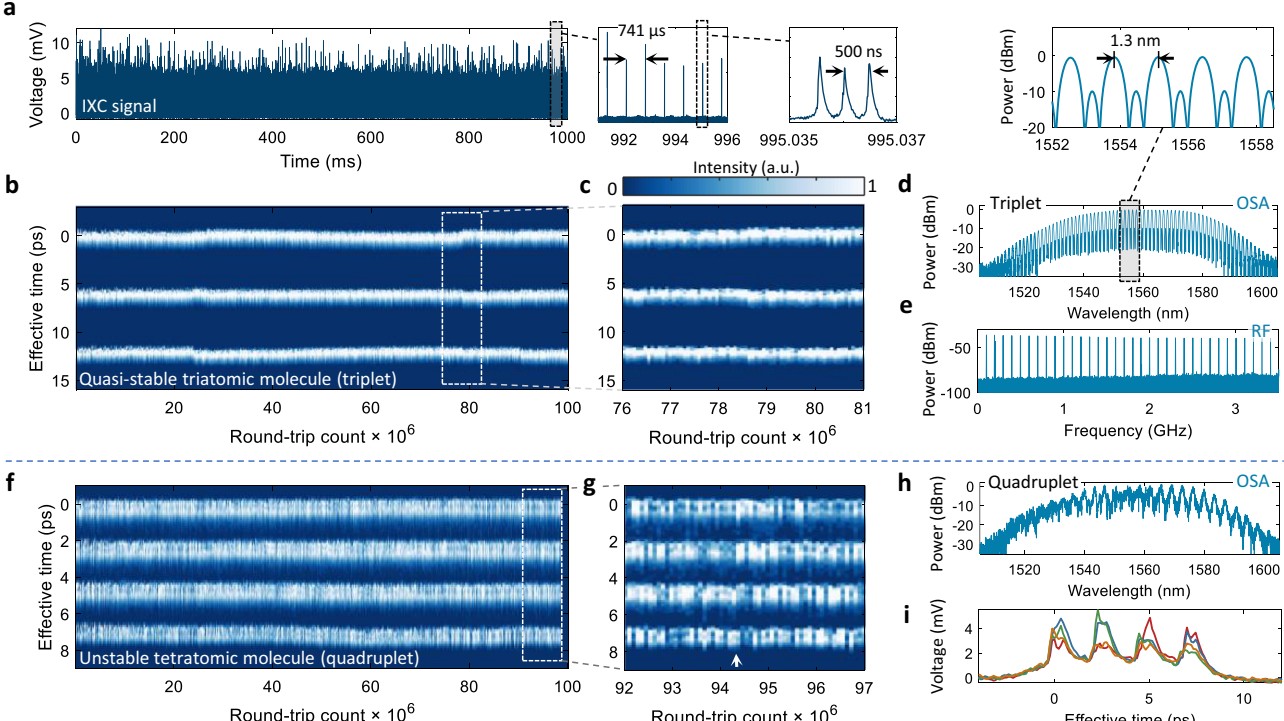

**Fig. 4 | IXC imaging of SM dynamics in a quasi-stable triatomic, and unstable tetratomic state. a** Amplified IXC signal acquired over 1 second, when the LUT was operated in the quasi-stable triatomic SM state. Subsequent panels show zoomed portions of the signal revealing the existence of 3 isolated pulses. **b** Imaging soliton molecule trajectory, where slight, sub-ps variations of the TS are visible. An analogously-long DFT waveform (1 s) measured using a 5 GHz 8-bit oscilloscope would have a size of ~10 GB, while the 12-bit 50 MHz bandwidth IXC dataset occupies 360 MB. Segmented acquisition in a narrower temporal span (to several microseconds each frame) may lower the size to ~1 MB. **c** Zoom of (**b**), which shows nearly-constant peak intensities. **d** Optical spectrum with a strong 1.2–1.3-nm-period modulation (~161 GHz) due to the presence of three bound pulses. **e** Radio-frequency (RF) spectrum measured by an InGaAs photodetector. It is nearly the same for panels (**f**–**i**). **f** IXC image of an unstable tetratomic molecule (quadruplet), with competition-like behavior between the pulses. **g** Zoom of the framed part showing the rapidly oscillating intensities of the SM pulses, in stark contrast to (**b**). **h** Optical spectrum acquired in parallel with IXC, which shows a dynamic evolution of the fringe contrast and noise-like envelope. **i** 4 slices of the IXC image in the region pointed by arrow in (**g**).

higher detector nonlinearities should unlock the IXC imaging potential of multi-GHz $f_r$ sources like microresonators with typical sub-pJ pulse energies and sub-W peak powers. Greater sensitivity obviously translates into better signal-to-noise (SNR) performance, particularly when the probed laser operates with mW-level average power (Fig. 6b).

## Discussion

Imaging of SM dynamics in a fiber laser cavity using a two-photon modality of DCS has been demonstrated. To achieve the high frame rates needed to capture the intramolecular motion over large optical bandwidths, two-photon IXC is measured. The technique enables one to surpass the conventional aliasing limit of DCS, while simultaneously offering eased requirements on the optical setup and large wavelength flexibility. We show that a mature 1550 nm mode-locked laser can probe another free-running mode-locked source operating at the same wavelength, or in the 1800–2100 nm wavelength range. This agility may be instrumental for diagnosing novel, mid-infrared mode-locked lasers[46], where the availability of optical components such as fibers or amplifiers for DFT is scarce. Even single-cavity dual-wavelength oscillators with asynchronous pulses that do not spectral overlap[47] can be directly studied for SM merely by illuminating a commercially-available two-photon photodiode.

IXC imaging of stable tri-atomic, and an unstable tetratomic SMs at kHz rates enables us to identify that the tetratomic molecule exhibits dynamic pulse competition effects, which corrupt the optical spectrum but are not directly visible in the RF spectrum. Probing this phenomenon at such rates and resolution would not have been possible using a swept OSA, conventional IAC or non-DFT-enhanced

oscilloscope. Next, increasing the repetition rate difference between the lasers to almost 100 kHz grants us access to high-rate imaging of a soliton rain, albeit at the expense of temporal resolution.

We believe that the IXC imaging technique fills the important niche between slow, time-averaged measurements performed by IAC or OSAs, and pulse-by-pulse diagnostics offered by DFT or direct pulse digitization by a high-bandwidth oscilloscope. From a practical standpoint, IXC is the perfect candidate for an easy-to-implement diagnostic tool in photonic laboratories to probe the existence of previously unseen loosely bound molecules due to the delay-resolution limit of autocorrelators or limited speed of oscilloscopes and photodiodes employed in DFT. For IXC-enabled laser diagnostics of sources with arbitrary repetition rates, we envision the application of amplified electro-optic (EOM) combs to serve as repetition-rate-agile probe sources.

Another strength of the IXC imaging technique lies in that it simplifies long-term SM studies while maintaining high scan rates. Because of the IXC signal's typically low duty cycle, segmented frame acquisition in a narrower effective time span may drastically reduce the amount of recorded data by orders of magnitude. This may prove to be difficult for competing spectral-domain techniques like DFT. Greatly benefit from IXC-enabled studies may also supercontinuum sources, whose large bandwidths make EFXC studies possible only at very low rates, while fiber parameters for implementing shot-to-shot DFT drastically change over the octave spans of supercontinua. Particularly such application scenarios call for developing two-photon detectors with higher optical nonlinearities to enable operation with lower average powers and broader optical pulses.

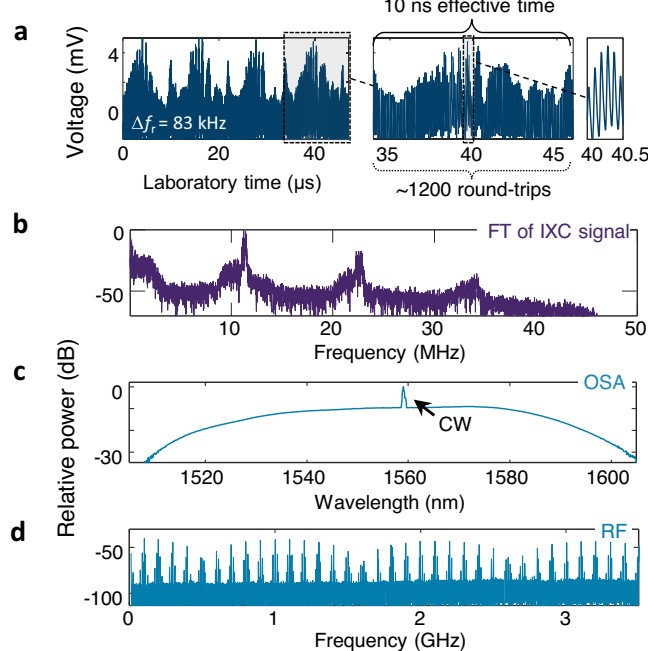

**Fig. 5 | Soliton-rain-like-event mediated by a strong CW component. a** IXC signal with $\Delta f_r = 83$ kHz scan rate, which is ~123 × above the aliasing limit. Zoomed panels resolve individual pulses. **b** Power spectrum of the IXC signal via the Fourier transform (FT). **c** Optical spectrum with a continuous-wave (CW) component. **d** Strongly modulated RF spectrum, which reflects the complex dynamics of this laser state at extended timescales.

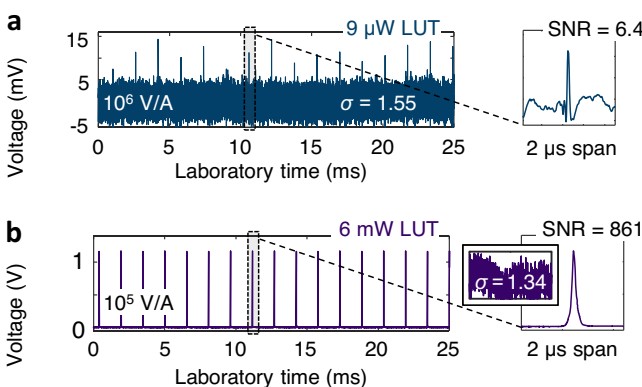

**Fig. 6 | InGaAsP quantum well laser as a two-photon photodetector. a** IXC signal in a soliton singlet state when the LUT average power was 9 μW. **b** IXC signal when the LUT average power was 6 mW. The LO power was 15 mW, and 23 mW, respectively.

## Methods

### Intensity cross-correlation

Asynchronous interaction between optical pulses on a photodetector that exhibits a square-law response with respect to intensity $I(t) = |E(t)|^2$ rather than $E$-field ($P(t) \sim I(t)^2$) produces an intensity cross-correlation (IXC) signal. The instantaneous optical power seen by the two-photon-absorption (2PA) detector is

$$P_{2PA}(t) = I_1^2(t) + I_1(t)I_2(t) + I_2^2(t). \tag{1}$$

Extended timescale averaging / integration yields a photodetector voltage

$$V \sim \langle P_{2PA}(t) \rangle = \frac{1}{2}\left( \int_{-\infty}^{\infty} I_1(t)^2 \, dt + \int_{-\infty}^{\infty} I_2(t)^2 \, dt + \int_{-\infty}^{\infty} I_1(t)I_2(t) \, dt \right). \tag{2}$$

Adding a variable time delay $\tau$ (lag) between $I_1$ and $I_2$ makes this signal more informative. The photodetector response now reads:

$$V(\tau) \sim \langle P_{2PA}(t), \tau \rangle = \frac{1}{2}\left( \langle I_1^2(t) \rangle + \langle I_2^2(t+\tau) \rangle + \langle I_1(t)I_2(t+\tau) \rangle \right). \tag{3}$$

One recognizes the intensity cross-correlation in the last term in addition to near-DC time-averaged squared pulse energies $I_{1,2}^2(t)$. This is the relevant term that survives in the electrical signal after low-pass filtering at $f_r/2$:

$$\langle I_1(t)I_2(t+\tau) \rangle = \int_{-\infty}^{\infty} I_1(t)I_2(t+\tau) \, dt = I_1(t) \star I_2(t). \tag{4}$$

In its most straightforward realization, measuring the IXC requires a mechanical stage that introduces lag $\tau = 2\Delta L/c$ when moved by $\Delta L$ (thus bounding the domain of integration of the improper cross-correlation integral). Unfortunately, the mechanical nature of the scan makes it obviously slow and suffers from resolution limitations analogous to Fourier Transform Spectroscopy (FTS). High resolution studies simply require long optical displacements. These limitations can be circumvented if one restricts the class of sources under study to pulsed lasers, or more generally sources with a well-defined repetition rate $f_r$. For such, an elegant way to introduce a variable lag is provided by means of dual-comb spectroscopy (DCS)[31,48] or equivalently asynchronous optical sampling (ASOPS)[49]. A second repetition-rate-mismatched source ($f_r + \Delta f_r$) provides a linearly varying (although discretized) lag that increases by $\Delta T \approx \Delta f_r/f_r^2$ from pulse pair to pulse pair[42] (Fig. 1b). This periodically samples (cross-correlates) the waveform over optical delays between 0 and $1/f_r$ thus producing an intensity cross-correlogram stretched in time by $m = f_r/\Delta f_r$. The process repeats itself at a $1/\Delta f_r$ rate, which ranges between nanoseconds to sub-seconds.

### Comparison with EFXC

Due to differences between IXC and conventional EFXC/DCS, we provide Table 1 that summarizes the prerequisites and limitations of the two techniques. This comparison is valid for a laser repetition frequency of $f_r = 100$ MHz or lower. Higher repetition rates (GHz) imply lower peak intensities, which necessitates more average power for IXC detection.

### Violation of the Nyquist criterion

To provide a mathematical formulation of violating the conventional (DCS) Nyquist limit, we will consider a linearly chirped Gaussian pulse with a spectral width $\Delta \nu$, which closely approximates a sech² optical pulse encountered in many laser systems[50]. The pulse has an electric field

$$E(t) = e^{-at^2} e^{i(\omega_0 t + bt^2)} \tag{5}$$

and intensity

$$I(t) = |E(t)|^2 = e^{-2at^2} = e^{-4\ln 2(t/\tau_p)^2}, \tag{6}$$

**Table 1 | Comparison of dual-comb based techniques for imaging of SM dynamics and laser diagnostics**

|  | EFXC | IXC |
|---|---|---|
| Response | Linear | Nonlinear (two-photon) |
| Interferometric | Yes | No |
| Polarization matching required | Yes | No[§] |
| Required average power | 1–100 µW | ~10 µW*–100 mW |
| Required pulse-width | Arbitrary | ~ps[¶] or lower |
| Spectral overlap | Required | No, only photon energies must add |
| Access to molecular phase | Yes | No |
| Lock between the lasers | Not needed for multiplexed cavities | Not needed |
| Aliasing problem | Yes, bandwidth $\Delta\nu \leq f_r^2/(2\Delta f_r)$ | No[†], but limited by the LUT dynamics |

*EFXC* electric field cross-correlation, *IXC* intensity cross-correlation.

[§]Avoiding polarization alignment between the two lasers prevents interferometric effects that would arise like in fringe-resolved IAC when operating at the same wavelength.

*Photodetectors with higher nonlinear coefficients should grant access to probing pulsed lasers with 1 µW or lower average power. Currently, it is several µW.

[¶]Longer pulses (dozens of ps) should also be detectable when a higher sensitivity two-photon photodetector is used.

[†]Operation high above the aliasing limit requires a careful choice of $\Delta f_r$ with respect to the pulse width.

where $a$ relates to the pulse's full width at half maximum (FWHM) $\tau_p$ through

$$\tau_p = \sqrt{\frac{2\ln 2}{a}}. \tag{7}$$

In Eq. (5), $b$ is the chirp parameter defining the sweep rate of the instantaneous frequency: $\omega(t) = \omega_0 + bt$ introduced by dispersion.

In conventional DCS, the (mutual) spectral width between the LO and LUT ultimately defines the maximum frame rate and hence $\Delta f_r$. The presence of chirp matters little here from an aliasing standpoint. Assuming both spectral widths $\Delta\nu$ being equal, the conventional aliasing limit is

$$\Delta f_r \leq \frac{f_r^2}{2\Delta\nu}. \tag{8}$$

This, however, greatly changes for two-photon detection. Because of chirp, the optical spectral width $\Delta\nu$ no longer implies a pulse width. Instead, in only imposes a lower bound on $\tau_p$ known as the Fourier transform limit:

$$\tau_d \geq \frac{2\ln 2}{\pi\Delta\nu} \approx \frac{0.44}{\Delta\nu}. \tag{9}$$

In other words, the presence of chirp leading to pulse broadening to duration $\tau_d$ can be seen as equivalent to optical band-pass filtering in conventional DCS. It reduces the occupied electrical bandwidth $B$ in the measured signal by

$$\frac{B_{\text{IXC}}}{B_{\text{EFXC}}} = \frac{1}{\sqrt{1 + \left(\frac{b}{a}\right)^2}}. \tag{10}$$

Propagation of a transform-limited pulse through a medium with group delay dispersion $D_2$ causes the pulse to elongate according to

$$\tau_d = \tau_p \sqrt{1 + \left(4\ln 2\frac{D_2}{\tau_p^2}\right)^2}. \tag{11}$$

Consequently, the conventional aliasing condition due to the Fourier pulse width limit

$$\Delta f_{r,p} \leq \frac{\pi}{4\ln 2}f_r^2\tau_p \approx 1.33 f_r^2\tau_p \tag{12}$$

is replaced in IXC with

$$\Delta f_{r,d} \leq \frac{\pi}{4\ln 2}f_r^2\tau_d. \tag{13}$$

From a practical standpoint, chirping the pulse is easier in implementation than selective optical band-pass filtering. It should be noted, however, that this operation lowers the obtainable temporal resolution in IXC studies, as discussed in the main text.

To better illustrate this effect, we will provide a numerical example. Consider a 60-nm wide optical spectrum centered at 1550 nm emitted by a laser with $f_r = 100$ MHz, similar to that presented here ($\Delta\nu \approx 7.5$ THz). The Fourier-limited Gaussian pulse width is ~ 68 fs. Using an equally-broad LO comb in conventional DCS implies $\Delta f_r \leq 667$ Hz. However, guiding the pulse through a 55-cm long piece of single-mode fiber (PM-1550 XP) ($D_2 \approx -1.2 \cdot 10^4\,\text{fs}^2$) increases its duration to 0.5 ps. This in turn relaxes the IXC Nyquist criterion ~ 7 times, yielding the maximum unaliased $\Delta f_r = 4.67$ kHz.

**Violation of the Nyquist criterion in the presence of prior assumptions**

Another reason why the IXC technique is more tolerant to operation above the Nyquist limit relates to the pulse observability criterion. If the IXC trace is to be used only for diagnostic purposes and not for accurate pulse width characterization, one can accept some degree of amplitude modulation of peak intensity no matter at what time instance it is detected. Note this is true for a quasi-stationary process, when the frame rate is much faster than pulse trajectory evolution. Accepting this apparent signal distortion due to amplitude underestimation can be seen as a form of controlled aliasing.

In the extreme case, we require the IXC signal to contain only one point when the pulses coincide (with some tolerable temporal offset that causes the modulation). This requires us to find a relative delay between the pulses that produces a signal weaker than for perfect overlap by factor $\eta$. Again, we will consider two Gaussian pulses with intensity profiles $I_1(t) = e^{-2at^2}$, and $I_2(t) = e^{-2bt^2}$. A cross-correlation ($\star$) of the two as a function of lag $\tau$, due to the shape symmetry, is equivalent to a convolution (*)

$$I_1 \star I_2 = \sqrt{\frac{\pi}{2(a+b)}}e^{-\frac{2ab}{a+b}\tau^2}. \tag{14}$$

This result can be derived from the central limit theorem – a convolution of two zero-mean Gaussians with variances $\sigma^2$ is a zero-mean Gaussian with its variance being the sum of the individual

variances $\sigma^2 = \sigma_1^2 + \sigma_2^2$. If we now ignore the proportionality constant, we define a new function $\eta(\tau)$, which provides a fraction of the maximum of the cross-correlation between the pulses at a given time lag. We will also relate it to the pulse widths $\tau_a$, $\tau_b$:

$$\eta(\tau) = e^{-\frac{2ab}{a+b}\tau^2} = 2^{-\frac{4}{\tau_a^2 + \tau_b^2}\tau^2}. \tag{15}$$

For two arbitrary amplitude underestimation factors: $\eta_{0.5} = \eta(\tau_{0.5}) = 0.5$, and $\eta_{0.1} = \eta(\tau_{0.1}) = 0.1$, we obtain the following analytical expressions

$$\tau_{0.5} = \pm \frac{1}{2} \sqrt{\tau_a^2 + \tau_b^2}, \tag{16}$$

$$\tau_{0.1} = \pm \frac{1}{2} \sqrt{\frac{\ln 10}{\ln 2}} \sqrt{\tau_a^2 + \tau_b^2} \approx 1.82\, \tau_{0.5}. \tag{17}$$

These quantities can be intuitively understood as the relative delay between the pulses that yields a given fraction (0.5 or 0.1) of the theoretical cross-correlation maximum at zero delay. Now we will recall that the asynchronous interaction of the pulses occurs at discrete time intervals increasing by $\Delta f_r / f_r^2$. Therefore, ensuring pulse detection (even in the case of a multi-pulse SM) with a relative amplitude not lower than $\eta$ requires the temporal separation to advance between consecutive cavity round-trips by

$$\frac{\Delta f_{r,\eta}}{f_r^2} \le 2\tau_\eta \rightarrow \Delta f_{r,\eta} \le 2\tau_\eta f_r^2. \tag{18}$$

The factor of 2 in $\tau_\eta$ relates to the fact that the pulse shape is symmetric, and either a positive or negative delay will produce the required response. For delays advancing by less than $\tau_\eta$, one would get two points above the predefined threshold. The condition defined in Eq. (18) works for pulse trajectory evolution occurring at rates much smaller than the frame rate $\Delta f_r$.

Again, to better illustrate these conditions, we will provide a numerical example. If we assume $f_r = 100$ MHz, and identical pulse widths $\tau_p = 500$ fs, we find that $\tau_{0.5} = \sqrt{2}\tau_p = 707.1$ fs, and $\tau_{0.1} = 1288.8$ fs. This corresponds to frame rates of $\Delta f_{r,0.5} \le 14.1$ kHz, and $\Delta f_{r,0.1} \le 25.8$ kHz, respectively, which exceed the conventional pulse-width-dependent aliasing limit 3 and 5.5 times, and the optical-bandwidth-dependent limit 21 and 38.5 times.

It is important to note that the Gaussian pulse shape is only an approximation. Sech$^2$ pulses, which are more difficult to treat analytically, have much longer wings, and even further relax the conventional aliasing condition, particularly in the $\tau_{0.1}$ case.

Another thing worth mentioning is that in the weakly-chirped regime, measuring the IXC high above the aliasing limit requires one to ensure an integer $k = f_r / \Delta f_r$ ratio to avoid signal scalloping, Vernier-like filtering effects. Because only discrete effective time intervals spaced by $1/(kf_r)$ are probed, features far away from this temporal grid may not be sampled, and hence overlooked. On the other hand, non-integer $k$ ratios offer enhanced temporal resolution due to scanning all possible LUT-probe relative delays like in a sampling oscilloscope. That said, acquiring such high-resolution scans takes multiple $\Delta f_r$ periods. Clearly, different trade-offs must be taken into account when violating the aliasing limit.

### Dual-wavelength IXC experiment
To obtain tunable pulses at longer wavelengths, we employ the soliton self-frequency shift (SSFS) effect in a highly nonlinear fiber[51]. Pulses from an oscillator are first amplified and compressed in an all-fiber erbium-doped fiber amplifier (EDFA) to dozens of mW of average power and sub-80 fs duration. Next, they are launched into a 2-m-long passive optical nonlinear fiber module for spectral shifting of solitons (Fibrain, Poland). By varying the pump current of the EDFA, we change the pulse peak power and thus the strength of the SSFS effect, which in turn governs the pulse center wavelength. To ensure that only the long-wave part of the spectrum interacts with the probe laser, we employ a cascade of wavelength domain multiplexing (WDM) filters/couplers to reject the original non-shifted pulse components (as seen in Fig. 2c) by at least 40 dB.

### Importance of the pulse amplifier
Because the intensity cross-correlogram is of pulsed unipolar nature, amplifying such a signal requires an amplifier with almost contradictory requirements: low noise figure, broad electrical bandwidth, and a low frequency limit extending almost to DC. Empirically, we found that the low frequency limit of 10 kHz is sufficient, while typical radio-frequency amplifiers operating above 1 MHz severely distort the IXC signal due to their high-pass properties. In all experiments here with the Si photodiode, we used a 2 GHz, 40 dB HSA-X-2-40 RF amplifier (Femto, Germany), which meets the aforementioned requirements. All IXC signal levels (in the mV range) indicated in this paper refer to the nominal signal from the unbiased Si photodiode after the 40 dB amplifier (100 V/V voltage gain).

### Fabry-Pérot diode laser as a two-photon detector
To demonstrate high-sensitivity two-photon detection using a commercial InGaAsP multi-quantum-well Fabry-Pérot diode laser, we coupled light into an NX7302AA-CC device (NEC) with a nominal emission wavelength of 1310 nm. For comparative purposes, the fiber length was kept the same as in the Si photodiode case. A Femto DHPCA-100 transimpedance amplifier boosted the signal from the detector with a gain of $10^6$ V/A and provided low-pass filtering to a cut-off frequency of 3.5 MHz. The minimum detectable average optical power (at a 6:1 voltage signal-to-noise ratio, peak to noise standard deviation) for a ~100 MHz source producing 0.5 ps long pulses was 9 μW.

### Optical power limitations
High average optical power can potentially damage the photodetector. In the case of the Si photodiode, illumination with a combined optical power of ~200 mW did not damage it (which was the highest optical power obtainable using our erbium-doped fiber amplifier). It is therefore uncertain, how much power the detector can withstand. Another issue that may arise in the experiment are limitations of the transimpedance amplifier. At high transimpedance gains ($10^6$ V/A), a clipping of the electrical signal has been observed when the incident optical power exceeded 10 mW for the FP laser detector. Therefore, a combination of moderate transimpedance gain with a high-vertical-resolution digitizer may be needed for optimal detection performance.

## Data availability
The data that support the plots within this paper and other findings of this study are available from the corresponding author upon request.

## Code availability
The mathematics and algorithms necessary to perform two-photon dual-comb imaging of soliton dynamics are described between the main text and Methods. The simple Matlab program used to plot the data from an oscilloscope is available from the corresponding author upon request.

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

## Acknowledgements

This project has received funding from the European Union's Horizon 2020 research and innovation programme under the Marie Skłodowska-Curie grant agreement (101027721, L.A.S.). This work is supported by the use of National Laboratory for Photonics and Quantum Technologies (NPLQT) infrastructure, which is financed by the European Funds under the Smart Growth Operational Programme. The authors also acknowledge funding from National Science Centre, Poland (2022/45/B/ST7/03316, J.S.). Dr. Jakub Boguslawski at Wroclaw University of Science and Technology, Poland is acknowledged for fruitful discussions on the importance of laser pulse diagnostics for nonlinear frequency conversion.

## Author contributions

L.A.S., and J.S. conceived the idea. L.A.S. carried out the optical and electrical measurements, analyzed the data, and generated the figures. J.S. fabricated the mode-locked lasers. L. A. S. wrote the manuscript with input from J. S. J.S. coordinated the project.

## Competing interests

The authors declare no competing interests.
