## [Peer Review File · Nature Communications]

Two-photon imaging of soliton dynamicsREVIEWER COMMENTS

Reviewer #1 (Remarks to the Author):

The authors describe a two-photon imaging method to study optical soliton dynamics. Specifically, the main benefit of using the two-photon absorption process is that the sampling pulse trains can be at a completely different wavelength from the probe pulse trains, which is of particular importance to studying optical solitons at longer wavelengths, like 2 μm . Also, the probe laser and the sampling laser do not need to be phase locked. In addition, unlike imaging methods using nonlinear crystal, the two-photon method does not require phase matching.

Overall the paper is well written, the idea is very novel, and it could have high impacts on nonlinear optics study. Therefore, I strongly recommend this paper for publication in Nature Communications.

I do have two small comments:

(1) In figure 1 (d), the frame rate for EFXC is listed as kHz. I understand this is accurate for low repetition rate pulsed laser. However, in microcavity [reference 3], the frame rate can be as high as 50 MHz. So I wonder if there is a better way to illustrate frame rate that includes both results in conventional mode-locked laser and the microcavity solitons.

(2) I don't really understand why the frame rate of the two-photon method can be higher than the optical coherent sampling. Is it because optical coherent sampling is phase sensitive and thus strictly limited by the Nyquist condition, while two-photon cross-correlation is not phase-sensitive and thus can violate the Nyquist condition to some extent? I think it will be better to clearly show what the frame rate limit is with a mathematical expression.

Reviewer #2 (Remarks to the Author):

In the paper titled "Two-photon imaging of soliton dynamics" by Łukasz A. Sterczewski et al., the authors demonstrate a way to imaging soliton molecules (SMs) based on non-interferometric intensity cross-correlation (IXC) technique. The local oscillator (1550nm) is a separate optical probe pulse stream generated at a repetition rate that is close to that of the SMs (1150-2200nm), and the small difference in these rates causes a pulse-to-pulse temporal shift of the probe pulses relative to the SM pulses. The coincidence of the two pulse streams on a photodetector with a large bandgap triggers two-photon absorption which does not require matching of wavelength or polarization. Evolutions of soliton molecules are captured as a proof-of-concept demonstration.

Overall, this mechanism is novel and should find applications in characterizing ultrafast pulses. I would recommend its publication in Nature Communications. Nevertheless, this technology also faces some practical challenges, which I request the authors to clarify in the revision.

1. Provided the low efficiency of the 2PA process, it is difficult to sample and reconstruct pulses with low pulse energy. Therefore, for high-rate pulses (like those generated in microresonators), this technology requires unpractically high average power that may damage the PD. The authors should evaluate the required pulse energy and repetition rate that gives a satisfactory SNR in their current setup.

2. Even if the average power of the pulse stream is kept low, aliasing of the detected signal would arise due to saturation of the photodetector induced by high peak powers of the pulses. This is a well-known limitation of dual-comb-based technologies. Please provide the actual power limitations of the detection scheme.

Besides, I have some questions:

1. The authors mentioned that " Since the ultimate goal is almost always high imaging speed to fully capture the SM evolution trajectory, ..., as shown in Fig. 3a. ". The analysis is not complete since the frame rates should exceed the evolution rate of the solitons. Otherwise, considerable details of soliton evolution would be lost.

2. The authors also mentioned that " In this context, worth studying is also a single-frame IXC temporal resolution limit, ...,the IXC signal due to the LPF lowers the peak contrast. ". Although $\Delta f_r/f_r^2$ does represent the ultimate limit of temporal resolution, the limit imposed by the pulse width of the local oscillator is also significant in many cases (especially when the difference between the repetition rates is small). This should be discussed in the main text.

3. In Figure 1d, the frame rate of EFXC can exceed 50 MHz with temporal resolution a few hundred femtoseconds [See Ref 3].

Reviewer #3 (Remarks to the Author):

The manuscript by L. A. Sterczewski and J. Sotor titled: "Two-photon imaging of soliton dynamics" reports stroboscopic two-photon imaging of the soliton molecules in a mode-locked laser. The authors demonstrate a simple method of detecting the shape of the laser solitons based on the intensity cross-correlation. More explicitly, they used two unsynchronized pulsed laser sources – one as a local oscillator and another one as a laser under test - operating at different wavelengths (1550 nm and 1800-2100 nm, respectively) that after a low pass filter were directed to a conventional photodiode operating at 400 - 1100 nm.

However, the publication of this manuscript in Nature Communication is not recommended for the following reasons:

1) The motivation of the study has several questionable statements.

a) About solitons in optics: "This is because optical solitons do not spread out during propagation and exhibit robustness against perturbations; therefore they frame the core concept in optical pulse generation". This statement does not fully reflect the motivation. Indeed, optics, first of all, provides an exceptional degree of control over the parameters and low propagation loss that made possible to generate solitons described by nearly integrable equations. This triggered the interest to create a soliton telecommunication line [1].

b) "Despite advances in mathematical modeling, the understanding of these complex inter-soliton interactions still appears to be in infancy." Even though it can be true for some advanced and complex systems, for the examples presented in the manuscript, the study of the formation of soliton molecules is definitely not in its infancy. Authors do not differentiate different platforms and put them into one basket (i) conservative solitons described by an integrable equation such as NLSE, (ii) dissipative solitons in a passive system such as microresonators governed by the Lugiato-Lefever equation, (iii) and dissipative solitons in an active system governed by the Ginzburg-Landau equation – the case experimentally investigated in the present manuscript. Soliton physics in these systems is very different. Indeed, in conservative systems, soliton molecules can be fully described analytically [2]. In passive resonators, this study of dissipative soliton interaction has been done in the 90s [3]. The literature on the soliton molecules and soliton interaction is extensive in the latter case as well [4].

c) One of the key motivation statements is: "To bypass these limitations and unlock the kHz- to sub-MHz rate imaging potential, in this Article we adapt the non-interferometric intensity cross-correlation". Implying that the dual-comb technique is incapable of achieving such rates, which contradicts the data presented in a table in the method section of a recent paper by Caldwell et. al. [5]

2) Authors do not discuss several powerful ultrafast measurement techniques, such as temporal imaging and its extensions [6]. These techniques have been employed not only for the single-shot detection of the temporal profile [7] but also for the full field characterization [8]. The last one is

the prominent example of the study of soliton (as well as solitons ensembles) build-up in a passively mode-locked laser. Also, the possibility of implementing this technique in the free space optics, makes it insensitive to the fiber transparency window [9].

3) The scheme is simple yet powerful. However, it is a natural extension of previously known techniques which obscures the novelty of the research. This method is a superposition of the conventional nonlinear cross-correlation technique extended by the TPA. As a result, very similar experimental techniques have been proposed only a few years after the discovery of the first laser, in 1968 by M. A. Duguay and J. W. Hansen [10]. Also, a similar approach using TPA has been used in spectroscopy [11].

4) Importantly, the phase reconstruction – in contrast to the paper that inspired this research (Ref. [29] of the manuscript) – is not shown in the manuscript, which makes it of limited interest to the community.

5) The same concerns the laser soliton dynamics. Effects described there have been reported and observed previously, mainly using DFT [12].

Minor comment: a large number of unnecessary and unconventional acronyms makes the paper difficult to read.

Concluding, the manuscript presents a study of a well-known problem using an original but anticipated technique which is a slight modification of well-known results. Thus, I confirm that the paper is publishable in a scientific journal but does not meet the novelty criteria in this particular case.

1. A. Hasegawa, "Soliton-Based Optical Communications: An Overview," *IEEE Journal of Selected Topics in Quantum Electronics* 6, no. 6 (November 2000): 1161–72, <https://doi.org/10.1109/2944.902164>.
2. Gang Xu et al., "Breather Wave Molecules," *Physical Review Letters* 122, no. 8 (February 27, 2019): 084101, <https://doi.org/10.1103/PhysRevLett.122.084101>.
3. S. Wabnitz, "Suppression of Interactions in a Phase-Locked Soliton Optical Memory," *Optics Letters* 18, no. 8 (April 15, 1993): 601, <https://doi.org/10.1364/OL.18.000601>.
4. Philippe Grelu and Nail Akhmediev, "Dissipative Solitons for Mode-Locked Lasers," *Nature Photonics* 6, no. 2 (February 2012): 84–92, <https://doi.org/10.1038/nphoton.2011.345>.
5. Emily D. Caldwell et al., "The Time-Programmable Frequency Comb and Its Use in Quantum-Limited Ranging," *Nature* 610, no. 7933 (October 27, 2022): 667–73, <https://doi.org/10.1038/s41586-022-05225-8>.
6. Brian H. Kolner and Moshe Nazarathy, "Temporal Imaging with a Time Lens," *Optics Letters* 14, no. 12 (June 15, 1989): 630–32, <https://doi.org/10.1364/OL.14.000630>.
7. Mikko Närhi et al., "Real-Time Measurements of Spontaneous Breathers and Rogue Wave Events in Optical Fibre Modulation Instability," *Nature Communications* 7, no. 1 (December 19, 2016): 13675, <https://doi.org/10.1038/ncomms13675>.
8. P. Ryczkowski et al., "Real-Time Full-Field Characterization of Transient Dissipative Soliton Dynamics in a Mode-Locked Laser," *Nature Photonics* 12, no. 4 (April 2018): 221–27, <https://doi.org/10.1038/s41566-018-0106-7>.
9. Pierre Suret et al., "Single-Shot Observation of Optical Rogue Waves in Integrable Turbulence Using Time Microscopy," *Nature Communications* 7, no. 1 (October 7, 2016): 13136, <https://doi.org/10.1038/ncomms13136>.
10. M. A. Duguay and J. W. Hansen, "Optical Sampling of Subnanosecond Light Pulses," *Applied Physics Letters* 13, no. 5 (September 1968): 178–80, <https://doi.org/10.1063/1.1652560>.
11. Marcus Rasmusson et al., "On the Use of Two-Photon Absorption for Determination of Femtosecond Pump-Probe Cross-Correlation Functions," *Chemical Physics Letters* 335, no. 3 (February 23, 2001): 201–8, [https://doi.org/10.1016/S0009-2614\(01\)00057-4](https://doi.org/10.1016/S0009-2614(01)00057-4).
12. Junsong Peng et al., "Real-Time Observation of Dissipative Soliton Formation in Nonlinear Polarization Rotation Mode-Locked Fibre Lasers," *Communications Physics* 1, no. 1 (December 2018): 20, <https://doi.org/10.1038/s42005-018-0022-7>; G. Herink et al., "Real-Time Spectral Interferometry Probes the Internal Dynamics of Femtosecond Soliton Molecules," *Science* 356, no. 6333 (April 7, 2017): 50–54, <https://doi.org/10.1126/science.aal5326>; Xueming Liu, Xiankun Yao, and Yudong Cui, "Real-Time Observation of the Buildup of Soliton Molecules," *Physical Review Letters* 121, no. 2 (July 12, 2018): 023905, <https://doi.org/10.1103/PhysRevLett.121.023905>.

Dear Editor, dear Reviewers,

We would like to thank the Reviewers for providing insightful comments and suggestions that helped us strengthen the manuscript. Below, are the detailed changes in the manuscript addressing the Reviewers' comments (our responses are shown in *blue*).

Reviewer #1 (Remarks to the Author):

The authors describe a two-photon imaging method to study optical soliton dynamics. Specifically, the main benefit of using the two-photon absorption process is that the sampling pulse trains can be at a completely different wavelength from the probe pulse trains, which is of particular importance to studying optical solitons at longer wavelengths, like 2 μm . Also, the probe laser and the sampling laser do not need to be phase locked. In addition, unlike imaging methods using nonlinear crystal, the two-photon method does not require phase matching.

Overall the paper is well written, the idea is very novel, and it could have high impacts on nonlinear optics study. Therefore, I strongly recommend this paper for publication in *Nature Communications*.

I do have two small comments:

(1) In figure 1 (d), the frame rate for EFXC is listed as kHz. I understand this is accurate for low repetition rate pulsed laser. However, in microcavity [reference 3], the frame rate can be as high as 50 MHz. So I wonder if there is a better way to illustrate frame rate that includes both results in conventional mode-locked laser and the microcavity solitons.

We fully agree with the need to better illustrate the obtainable frame rate Δf_r irrespective of the repetition rate f_r . One of the big difficulties is that permissible Δf_r does not scale linearly with the repetition rate, and additionally depends on the probed optical bandwidth. To satisfy the Reviewer's requirement, we propose a figure-of-merit (FOM) in units of Hz^2/Hz^2 derived from the aliasing criterion.

In conventional EFXC, the frame rate must meet the condition $\Delta f_r \leq f_r^2/2\Delta\nu$, where $\Delta\nu$ is the probed optical bandwidth (practically defined for a -20 dB drop in intensity). To describe the "speed" of the technique we propose to calculate how much above the Nyquist criterion (yet only for sampling techniques, to which DFT does not belong) one can image the soliton trajectory.

In conventional EFXC the condition is obviously

$$2\Delta\nu\Delta f_r / f_r^2 \leq 1 = \text{FOM}_{\text{EFXC}}.$$

For instance, plugging data from Ref. 3, $\Delta\nu = 1.625$ THz, $f_r = 22$ GHz, $\Delta f_r = 50$ MHz, provides a FOM of 0.34. This means that imaging was performed at $\sim 1/3$ of the (optical) Nyquist-limited acquisition speed.

In the case of the setup presented here ($\Delta\nu = 7.5$ THz, $f_r = 100.8$ MHz, $\Delta f_r = 670$ Hz for a theoretical Nyquist-limited frame rate), we obtain an FOM of 0.99 if the experiment were performed in EFXC mode. However, in the above-Nyquist limit presented in the paper, when Δf_r varies between 1 kHz and 100 kHz, the corresponding FOMs are 2.11, and 147.6. This is possible because the FOM must no longer be lower or equal to one. Of course, such high rates come at the expense of lowered temporal

resolution, in which case distinguishing closely spaced soliton pulses may be difficult or even impossible.

For non-sampling techniques, the obtainable frame rate reaches the laser repetition rate $\Delta f_r = f_r$. Then, the FOM approaches infinity as the bandwidth increases.

$$2\Delta\nu / f_r = \text{FOM}_{\text{NS}}$$

We believe this addresses the requirement for dimensionless comparison of the obtainable frame rates. The above discussion has been added to the caption of Fig. 1.

(2) I don't really understand why the frame rate of the two-photon method can be higher than the optical coherent sampling. Is it because optical coherent sampling is phase sensitive and thus strictly limited by the Nyquist condition, while two-photon cross-correlation is not phase-sensitive and thus can violate the Nyquist condition to some extent? I think it will be better to clearly show what the frame rate limit is with a mathematical expression.

Per Reviewer's request, we have added a page-long mathematical formulation of how the conventional Nyquist criterion can be violated by means of dispersion (pulse elongation given a fixed optical bandwidth), and prior assumptions about the pulse. Please see sections Violation of the Nyquist criterion, and Violation of the Nyquist criterion in the presence of prior assumptions in the Methods section for details. For Reviewer's convenience it is also provided below:

Violation of the Nyquist criterion.

To provide a mathematical formulation of violating the conventional (DCS) Nyquist limit, we will consider a linearly chirped Gaussian pulse with a spectral width $\Delta\nu$, which closely approximates a sech^2 optical pulse encountered in many laser systems. The pulse has an electric field

$$E(t) = e^{-at^2} e^{i(\omega_0 t + bt^2)}. \quad (5)$$

and intensity

$$I(t) = |E(t)|^2 = e^{-2at^2} = e^{-4 \ln 2 (t/\tau_p)^2}. \quad (6)$$

where a relates to the pulse's full width at half maximum (FWHM) τ_p through

$$\tau_p = \sqrt{\frac{2 \ln 2}{a}}. \quad (7)$$

In Eq.5, b is the chirp parameter defining the sweep rate of the instantaneous frequency: $\omega(t) = \omega_0 + bt$ introduced by dispersion.

In conventional DCS, the (mutual) spectral width between the LO and LUT ultimately defines the maximum frame rate and hence Δf_r . The presence of chirp matters little here from an aliasing standpoint. Assuming both spectral widths $\Delta\nu$ being equal, the conventional aliasing limit is

$$\Delta f_r \leq \frac{f_r^2}{2\Delta\nu}. \quad (8)$$

This, however, greatly changes for two-photon detection. Because of chirp, the optical spectral width $\Delta\nu$ no longer implies a pulse width. Instead, it only imposes a lower bound on τ_p known as the Fourier transform limit:

$$\tau_d \geq \frac{2 \ln 2}{\pi \Delta \nu} \approx \frac{0.44}{\Delta \nu}. \quad (9)$$

In other words, the presence of chirp leading to pulse broadening to duration τ_d can be seen as equivalent to optical band-pass filtering in conventional DCS. It reduces the occupied electrical bandwidth B in the measured signal by

$$\frac{B_{\text{IXC}}}{B_{\text{EFXC}}} = \frac{1}{\sqrt{1 + \left(\frac{b}{a}\right)^2}}. \quad (10)$$

Propagation of a transform-limited pulse through a medium with group delay dispersion D_2 causes the pulse to elongate according to

$$\tau_d = \tau_p \sqrt{1 + \left(4 \ln 2 \frac{D_2}{\tau_p^2}\right)^2}. \quad (11)$$

Consequently, the conventional aliasing condition due to the Fourier pulse width limit

$$\Delta f_{r,p} \leq \frac{\pi}{4 \ln 2} f_r^2 \tau_p \approx 1.33 f_r^2 \tau_p \quad (12)$$

is replaced in IXC with

$$\Delta f_{r,d} \leq \frac{\pi}{4 \ln 2} f_r^2 \tau_d. \quad (13)$$

From a practical standpoint, chirping the pulse is easier in implementation than selective optical band-pass filtering. It should be noted, however, that this operation lowers the obtainable temporal resolution in IXC studies.

To better illustrate this effect, we will provide a numerical example. Consider a 60-nm wide optical spectrum centered at 1550 nm emitted by a laser with $f_r = 100$ MHz, similar to that presented here ($\Delta \nu \approx 7.5$ THz). The Fourier-limited Gaussian pulse width is ~ 68 fs. Using an equally-broad LO comb in conventional DCS implies $\Delta f_r \leq 667$ Hz. However, guiding the pulse through a 55-cm long piece of single-mode fiber (PM-1550 XP) ($D_2 \approx -1.2 \times 10^4$ fs²) increases its duration to 0.5 ps. This in turn relaxes the IXC Nyquist criterion ~ 7 times, yielding the maximum unaliased $\Delta f_r = 4.67$ kHz.

Violation of the Nyquist criterion in the presence of prior assumptions.

Another reason why the IXC technique is more tolerant to operation above the Nyquist limit relates to the pulse observability criterion. If the IXC trace is to be used only for diagnostic purposes and not for accurate pulse width characterization, one can accept some degree of amplitude modulation of peak intensity no matter at what time instance it is detected. Note this is true for a quasi-stationary process, when the frame rate is much faster than pulse trajectory evolution. Accepting this apparent signal distortion due to amplitude underestimation can be seen as a form of controlled aliasing.

In the extreme case, we require the IXC signal to contain only one point when the pulses coincide (with some tolerable temporal offset that causes the modulation). This requires us to find a relative delay between the pulses that produces a signal weaker than for perfect overlap by factor η . Again, we will consider two Gaussian pulses with intensity profiles $I_1(t) = \exp(-2at^2)$, and $I_2(t) = \exp(-2bt^2)$. A cross-correlation (\star) of the two as a function of lag τ , due to the shape symmetry, is equivalent to a convolution (\ast)

$$I_1 \star I_2 = \sqrt{\frac{\pi}{2(a+b)}} e^{-\frac{2ab}{a+b}\tau^2}. \quad (14)$$

This result can be derived from the central limit theorem -- a convolution of two zero-mean Gaussians with variances σ^2 is a zero-mean Gaussian with its variance being the sum of the individual variances $\sigma^2 = \sigma_1^2 + \sigma_2^2$. If we now ignore the proportionality constant, we define a new function $\eta(\tau)$, which provides a fraction of the maximum of the cross-correlation between the pulses at a given time lag. We will also relate it to the pulse widths τ_a, τ_b :

$$\eta(\tau) = e^{-\frac{2ab}{a+b}\tau^2} = 2^{-\frac{4}{\tau_a^2 + \tau_b^2}\tau^2}. \quad (15)$$

For two arbitrary amplitude underestimation factors: $\eta_{0.5} = \eta(\tau_{0.5}) = 0.5$, and $\eta_{0.1} = \eta(\tau_{0.1}) = 0.1$, we obtain the following analytical expressions

$$\tau_{0.5} = \pm \frac{1}{2} \sqrt{\tau_a^2 + \tau_b^2}, \quad (16)$$

$$\tau_{0.1} = \pm \frac{1}{2} \sqrt{\frac{\ln 10}{\ln 2}} \sqrt{\tau_a^2 + \tau_b^2} \approx 1.82 \tau_{0.5}. \quad (17)$$

These quantities can be intuitively understood as the relative delay between the pulses that yields a given fraction (0.5 or 0.1) of the theoretical cross-correlation maximum at zero delay. Now we will recall that the asynchronous interaction of the pulses occurs at discrete time intervals increasing by $\Delta f_r / f_r^2$. Therefore, ensuring pulse detection (even in the case of a multi-pulse SM) with a relative amplitude not lower than η requires the temporal separation to advance between consecutive cavity round-trips by

$$\frac{\Delta f_{r,\eta}}{f_r^2} \leq 2\tau_\eta \rightarrow \Delta f_{r,\eta} \leq 2\tau_\eta f_r^2. \quad (18)$$

The factor of 2 in τ_η relates to the fact that the pulse shape is symmetric, and either a positive or negative delay will produce the required response. For delays advancing by less than τ_η , one would get two points above the predefined threshold. The condition defined in Eq. 18 works for pulse trajectory evolution occurring at rates much smaller than the frame rate Δf_r .

Again, to better illustrate these conditions, we will provide a numerical example. If we assume $f_r = 100$ MHz, and identical pulse widths $\tau_p = 500$ fs, we find that $\tau_{0.5} = \sqrt{2}\tau_p = 707.1$ fs, and $\tau_{0.1} = 1288.8$ fs. This corresponds to frame rates of $\Delta f_{r,0.5} \leq 14.1$ kHz, and $\Delta f_{r,0.1} \leq 25.8$ kHz, respectively, which exceed the conventional pulse-width-dependent aliasing limit 3 and 5.5 times, and the optical-bandwidth-dependent limit 21 and 38.5 times.

It is important to note that the Gaussian pulse shape is only an approximation. Sech² pulses, which are more difficult to treat analytically, have much longer wings, and even further relax the conventional aliasing condition, particularly in the $\tau_{0.1}$ case.

Another thing worth mentioning is that in the weakly-chirped regime, measuring the IXC high above the aliasing limit requires one to ensure an integer $k = f_r / \Delta f_r$ ratio to avoid signal scalloping, Vernier-like filtering effects. Because only discrete effective time intervals spaced by $1/(kf_r)$ are probed, features far away from this temporal grid may not be sampled, and hence overlooked. On the other hand, non-integer k ratios offer enhanced temporal resolution due to scanning all possible LUT-probe relative delays like in a sampling oscilloscope. That said, acquiring such high-resolution scans takes multiple Δf_r periods. Clearly, different trade-offs must be taken into account when violating the aliasing limit.

Reviewer #2 (Remarks to the Author):

In the paper titled “Two-photon imaging of soliton dynamics” by Łukasz A. Sterczewski et al., the authors demonstrate a way to imaging soliton molecules (SMs) based on non-interferometric intensity cross-correlation (IXC) technique. The local oscillator (1550nm) is a separate optical probe pulse stream generated at a repetition rate that is close to that of the SMs (1150-2200nm), and the small difference in these rates causes a pulse-to-pulse temporal shift of the probe pulses relative to the SM pulses. The coincidence of the two pulse streams on a photodetector with a large bandgap triggers two-photon absorption which does not require matching of wavelength or polarization. Evolutions of soliton molecules are captured as a proof-of-concept demonstration.

Overall, this mechanism is novel and should find applications in characterizing ultrafast pulses. I would recommend its publication in Nature Communications. Nevertheless, this technology also faces some practical challenges, which I request the authors to clarify in the revision.

1. Provided the low efficiency of the 2PA process, it is difficult to sample and reconstruct pulses with low pulse energy. Therefore, for high-rate pulses (like those generated in microresonators), this technology requires unpractically high average power that may damage the PD. The authors should evaluate the required pulse energy and repetition rate that gives a satisfactory SNR in their current setup.

We are thankful for this suggestion as it has encouraged us to search for more sensitive 2-photon detectors and explore optical power constraints of the current Si detector.

First, we found that a combined optical average power as high as 200 mW does not damage the silicon photodiode. It was the highest power we could obtained using our erbium doped fiber amplifier. After decreasing the power to the original level, the detector responds identically as prior to the high-power illumination.

However, what we find much more appealing to the readers, and potentially to the Reviewer is that using an InGaAsP 1.3 μm Fabry-Perot telecom laser diode structure (operating here as a 2PA detector see Reference 42, listed below), we have successfully probed 1.5 μm laser pulses using the IXC technique at LUT power levels as low as 9 μW . This represents a sensitivity improvement by three orders of magnitude. The only downside is that in the current implementation, a short piece of fiber attached to the laser adds some dispersion, yet will be eliminated in a future free-space setup and is not a limitation of the technique per se. The following paragraph along with Fig. 6 have been added to satisfy the Reviewer’s comment and reflect this major addition to the manuscript.

Although the Si photodiode requires watts of peak optical power to produce a detectable signal (sub-nJ pulse energy for sub-ps durations), much higher sensitivity two-photon detectors can be used instead. For instance, quantum-well laser structures have proven to exhibit 2PA sensitivities orders of magnitude higher than semiconductor photodiodes (42). For the 1550 nm range, excellent performance is offered by multi-quantum-well Fabry-Perot laser diodes with nominal emission wavelengths at 1.3 μm , as shown in Fig. 6 (see Methods for device details). Whereas for the Si photodiode the sensitivity defined as a product of the peak LUT power and average LO power (42) amounts to $\sim 4.4 \times 10^6 \text{ mW}^2$ using an unbiased laser diode as a detector yields an improvement to $2.4 \times 10^3 \text{ mW}^2$. In other words, we can probe μW average power level pulses instead of mW with corresponding fJ pulse energies. This is shown in Fig. 6a, where 9 μW , 0.5 ps-long pulses are probed by a 15 mW LO. This major sensitivity improvement offered by higher detector nonlinearities should unlock the IXC imaging potential of multi-

GHz f_r sources like microresonators with typical sub-pJ pulse energies and sub-W peak powers. Greater sensitivity obviously translates into better signal-to-noise (SNR) performance, particularly when the probed laser operates with mW-level average power (Fig. 6b).

Fig. 6 - InGaAsP quantum well laser as a two-photon photodetector. a IXC signal in a soliton singlet state when the LUT average power was 9 μW . **b** IXC signal when the LUT average power was 6 mW. The LO power was 15 mW, and 23 mW, respectively.

(42): Reid, D. et al. *Commercial semiconductor devices for two photon absorption autocorrelation of ultrashort light pulses. Applied optics* 37, 8142–8144 (1998)

Assuming that a pulsed laser with a repetition rate in the MHz range has a 1000 \times lower peak intensity than a GHz-rate laser at the same pulse duration, the improved detection sensitivity offered by laser diode structures should allow microresonators (or semiconductor laser frequency combs in general) to be probed using the IXC technique. It is only a question of a suitable two-photon detector. Our work may trigger efforts to push the 2PA sensitivity to new values.

2. Even if the average power of the pulse stream is kept low, aliasing of the detected signal would arise due to saturation of the photodetector induced by high peak powers of the pulses. This is a well-known limitation of dual-comb-based technologies. Please provide the actual power limitations of the detection scheme.

The reviewer has raised the important concern of photodetector saturation. Indeed, in conventional dual-comb spectroscopy optical powers as low as 100 μW can distort the interferogram and lead to improper absorption line shapes. Currently, special algorithms taking into account the generation of spectra at harmonic frequencies with respect to the carrier are being developed by the community to overcome this issue. (see Guay, Philippe, et al. "Correcting photodetector nonlinearity in dual-comb interferometry." *Optics Express* 29.18 (2021): 29165-29174., Guay, Philippe, et al. "Linear dual-comb interferometry at high power levels." *Optics Express* 31.3 (2023): 4393-4404.)

In the case of the two-photon detection process, we do not observe the clipping or distortion behavior of the cross-correlation signal as in dual-comb spectroscopy. What we find a much bigger concern is the necessity of using high-rejection-ratio optical long-pass filters (or a cascade of those). Residual light in the one-photon absorption range weakens the 2PA effect and produces a significant amount of noise. This obviously leads to a behavior different from the expected quadratic nonlinearity.

Nevertheless, we do observe saturation effects in the case of the Fabry-Perot diode detector, which, unexpectedly, are caused by electronics. At elevated optical powers (\sim dozens of mW), when the

transimpedance amplifier gain is set to 10^6 or higher, the amplifier clips the signal when it produces a volt-level signal at the output.

To satisfy the Reviewer's requirement, we have added the following to the Methods section:

High average optical power can potentially damage the photodetector. In the case of the Si photodiode, illumination with a combined optical power of ~200 mW did not damage it (which was the highest optical power obtainable using our erbium-doped fiber amplifier). It is therefore uncertain, how much power the detector can withstand. Another issue that may arise in the experiment are limitations of the transimpedance amplifier. At high transimpedance gains, a clipping of the electrical signal has been observed at when the incident optical power exceeded 10 mW for the FP laser detector. Therefore, a combination of moderate transimpedance gain with a high-vertical-resolution digitizer may be needed for optimal detection performance.

Besides, I have some questions:

1. The authors mentioned that “ Since the ultimate goal is almost always high imaging speed to fully capture the SM evolution trajectory, ..., as shown in Fig. 3a. ”. The analysis is not complete since the frame rates should exceed the evolution rate of the solitons. Otherwise, considerable details of soliton evolution would be lost.

The Reviewer is absolutely right that details of soliton evolution will be lost if the frame rate is too low. This problem is particularly challenging for typical fiber or solid-state lasers, which (as mentioned earlier) have a conventional Nyquist limit in the sub-kHz range corresponding to ~100 000 cavity round trips to produce a single frame. While the IXC technique will never reach the imaging speed of DFT or time lens because it does not operate a shot-to-shot basis, breaking the Nyquist limit tens or even a hundred times is already a major step towards capturing the rapid SM dynamics in such low-repetition-rate sources.

GHz-rate sources like microcavity resonators have the unique capability of being imaged at rates of 10's of MHz due to favorable scaling of imaging speed with a square of the repetition rate. In such scenarios, the IXC technique can offer probing sources in more challenging spectral regions while using mature telecom-grade optical modulators at 1.5 μm rather than a strong violation of the Nyquist criterion.

The manuscript now includes an extra sentence that talks about the incomplete picture of SM evolution at insufficiently high frame rates:

We need to underline, however, that if the soliton evolution rate exceeds the frame rate, one has to resort to single-shot imaging techniques like DTF (12), time-lens (26) or even a combination of both (28) to capture all details of the soliton evolution trajectory

References:

(12) Herink, G., Kurtz, F., Jalali, B., Solli, D. R. & Ropers, C. Real-time spectral interferometry probes the internal dynamics of femtosecond soliton molecules. *Science* 356, 50–54 (2017).

(26) Kolner, B. H. & Nazarathy, M. Temporal imaging with a time lens. *Optics letters* 14, 630–632 (1989).

(28) Ryzkowski, P. et al. Real-time full-field characterization of transient dissipative soliton dynamics in a mode-locked laser. *Nature Photonics* 12, 221–227 (2018).

2. The authors also mentioned that “ In this context, worth studying is also a single-frame IXC temporal resolution limit, ...,the IXC signal due to the LPF lowers the peak contrast. ”. Although $\Delta f_r/f_r^2$ does represent the ultimate limit of temporal resolution, the limit imposed by the pulse width of the local oscillator is also significant in many cases (especially when the difference between the repetition rates is small). This should be discussed in the main text.

We are thankful for this comment since the Reader may have a false perception of the obtainable resolution. For instance, despite a large temporal magnification factor, when the frame rate is low, probing features with tens of femtoseconds of width using a near-ps long LO pulse will smear out the details. Although in principle a deconvolution technique can be used to restore the original shape, it always comes at the expense of increased noise.

Just like in any sampling technique, the obtained temporal resolution will be the RMS sum of individual widths. This discussion has been now added to the text.

It should be also noted that the ultimate resolution limit of the technique, considering the LO pulse width σ_{LO} , and its jitter σ_j , should obey the root mean square sum law, i.e. the obtainable resolution will

$$be \sigma_t = \sqrt{\delta_t^2 + \sigma_{LO}^2 + \sigma_j^2}.$$

3. In Figure 1d, the frame rate of EFXC can exceed 50 MHz with temporal resolution a few hundred femtoseconds [See Ref 3].

We fully agree with the Reviewer. Please see our response to the same concern raised by Reviewer #1, Comment 1.

Reviewer #3 (Remarks to the Author):

The manuscript by L. A. Sterczewski and J. Sotor titled: “Two-photon imaging of soliton dynamics” reports stroboscopic two-photon imaging of the soliton molecules in a mode-locked laser. The authors demonstrate a simple method of detecting the shape of the laser solitons based on the intensity cross-correlation. More explicitly, they used two unsynchronized pulsed laser sources – one as a local oscillator and another one as a laser under test - operating at different wavelengths (1550 nm and 1800-2100 nm, respectively) that after a low pass filter were directed to a conventional photodiode operating at 400 - 1100 nm.

However, the publication of this manuscript in Nature Communication is not recommended for the following reasons:

1) The motivation of the study has several questionable statements.

a) About solitons in optics: “This is because optical solitons do not spread out during propagation and exhibit robustness against perturbations; therefore they frame the core concept in optical pulse generation”. This statement does not fully reflect the motivation. Indeed, optics, first of all, provides an exceptional degree of control over the parameters and low propagation loss that made possible to generate solitons described by nearly integrable equations. This triggered the interest to create a soliton telecommunication line [1].

b) “Despite advances in mathematical modeling, the understanding of these complex inter-soliton interactions still appears to be in infancy.” Even though it can be true for some advanced and complex systems, for the examples presented in the manuscript, the study of the formation of soliton molecules is definitely not in its infancy. Authors do not differentiate different platforms and put them into one

basket (i) conservative solitons described by an integrable equation such as NLSE, (ii) dissipative solitons in a passive system such as microresonators governed by the Lugiato-Lefever equation, (iii) and dissipative solitons in an active system governed by the Ginzburg-Landau equation – the case experimentally investigated in the present manuscript. Soliton physics in these systems is very different. Indeed, in conservative systems, soliton molecules can be fully described analytically [2]. In passive resonators, this study of dissipative soliton interaction has been done in the 90s [3]. The literature on the soliton molecules and soliton interaction is extensive in the latter case as well [4].

In our introduction we tried to encompass the diverse landscape of soliton phenomena. Obviously the picture we draw will always be subjective and will only attempt to briefly cover the exciting underlying physics given the manuscript word count constraints. In this work, we only attempt to draw a broader picture in the introduction, which obviously, may seem incomplete from a soliton physics expert's point of view.

Regarding the maturity of soliton molecule understanding, we tend to agree with the Reviewer that much work has been done in the early days of computer-aided numerical modeling. Nevertheless, novel soliton molecule states are still observed (despite the large number of available simulation packages) even in simple fiber resonators (i.e. molecular complexes or interaction between soliton molecules in different polarizations, as governed by the Complex Ginzburg-Landau equation).

c) One of the key motivation statements is: “To bypass these limitations and unlock the kHz- to sub-MHz rate imaging potential, in this Article we adapt the non-interferometric intensity cross-correlation”. Implying that the dual-comb technique is incapable of achieving such rates, which contradicts the data presented in a table in the method section of a recent paper by Caldwell et. al. [5]

In the manuscript we do not claim that interferometric cross-correlation is incapable of reaching such rates. In fact, the imaging rate is relative to the laser repetition rate, and scales with the square of it. Please see our response to Reviewer 1, point #1, where we highlight the difficulties that fiber lasers with MHz repetition rates face to be properly diagnosed/imaged.

2) Authors do not discuss several powerful ultrafast measurement techniques, such as temporal imaging and its extensions [6]. These techniques have been employed not only for the single-shot detection of the temporal profile [7] but also for the full field characterization [8]. The last one is the prominent example of the study of soliton (as well as solitons ensembles) build-up in a passively mode-locked laser. Also, the possibility of implementing this technique in the free space optics, makes it insensitive to the fiber transparency window [9].

It is impossible to disagree with the Reviewer that temporal imaging has been overlooked. This has been now discussed in the manuscript with suitable references.

Temporal imaging with a time lens (26) has also gained an established position as a tool for single-shot laser pulse diagnostics (27,28).

It should be noted, however, that the temporal imaging technique is not universal and often requires specialized equipment like phase modulators for the relevant spectral region or fibers with low four-wave mixing thresholds. Therefore, it is hampered by the same limitations as the DFT technique, albeit with the primary advantage of providing shot-to-shot laser diagnostics information. The mentioned free-space implementation of time lens requires a suitable nonlinear crystal (that requires phase matching) along with a pump laser at a completely different wavelength, which adds many layers of complexity compared to the fiber realization. As always, different trade-offs must be taken into account.

Still, we do inform the Reader that in some cases there is no other option but to use DFT or time lens:

We need to underline, however, that if the soliton evolution rate exceeds the frame rate, one has to resort to single-shot imaging techniques like DTF (12), time-lens (26) or even a combination of both (28) to capture all details of the soliton evolution trajectory.

References:

(12) Herink, G., Kurtz, F., Jalali, B., Solli, D. R. & Ropers, C. Real-time spectral interferometry probes the internal dynamics of femtosecond soliton molecules. Science 356, 50–54 (2017).

(26) Kolner, B. H. & Nazarathy, M. Temporal imaging with a time lens. Optics letters 14, 630–632 (1989).

(28) Ryzkowski, P. et al. Real-time full-field characterization of transient dissipative soliton dynamics in a mode-locked laser. Nature Photonics 12, 221–227 (2018).

3) The scheme is simple yet powerful. However, it is a natural extension of previously known techniques which obscures the novelty of the research. This method is a superposition of the conventional nonlinear cross-correlation technique extended by the TPA. As a result, very similar experimental techniques have been proposed only a few years after the discovery of the first laser, in 1968 by M. A. Duguay and J. W. Hansen [10]. Also, a similar approach using TPA has been used in spectroscopy [11].

In the manuscript we do not claim that the use of nonlinear cross-correlation is unprecedented. We properly acknowledge prior advances in the field. However, early demonstrations have explored nonlinear crystals that require phase matching, which may make sampling lasers at dissimilar wavelength impossible. Additionally, in addition to wavelength agility and polarization insensitivity, the technique offers probing femtojoule optical pulses, as shown in the revised manuscript per Reviewers' suggestions. It is therefore unfair to compare it with Ref. 11, which exploits mJ-level optical pulses with UV coverage to produce a cross-correlation signal.

4) Importantly, the phase reconstruction – in contrast to the paper that inspired this research (Ref. [29] of the manuscript) – is not shown in the manuscript, which makes it of limited interest to the community.

The intensity cross-correlation measurement in Ref. 29 resembles a conventional intensity autocorrelation setup with the notable difference that the laser pulse is analyzed by an unbalanced Michelson interferometer instead of a balanced one. One of the arms includes a dispersive element, which causes the measured cross-correlogram to become asymmetric and hence carry phase information. The same laser (only one source, not two) is measured twice, and a phase reconstruction technique can be used if the dispersion of the added element is known, and the optical spectrum is measured. The technique is referred to as PICASO.

This is in stark contrast to the much more difficult case of two optical pulses with unknown temporal profiles that produce an intensity cross-correlation signal as here. The number of unknown parameters to be estimated (and hence the solution space) is much larger. In principle, accompanying spectral measurements for each of the lasers, and a supplementary intensity autocorrelation of each of the interacting lasers can be used to guide the multidimensional optimization algorithm to reconstruct the phases. However, a suitable convergence analysis must be performed first. We are thankful for this suggestion as this sets new research directions for developing the technique. Nevertheless, given the problem complexity, which goes far beyond the single-source case, we believe that our

demonstrations still make it of practical relevance to the community by the same way the intensity autocorrelation plays an important role even without providing phase information.

5) The same concerns the laser soliton dynamics. Effects described there have been reported and observed previously, mainly using DFT [12].

This manuscript does not claim the discovery of new phenomena or soliton physics in fiber oscillators. In fact, the opposite is true: we image well-known or recently discovered SM states because such can be leveraged to better draw an analogy between existing pulse diagnostic techniques, and the wavelength-agile IXC technique. Agreement with prior studies only proves the validity of the idea.

Minor comment: a large number of unnecessary and unconventional acronyms makes the paper difficult to read.

We have revised the text to keep only relevant, well-established acronyms. An example of an unnecessary (and overused) acronym was PD, which now has been removed from the text. The same holds for IGM – the interferogram and many more. For convenience, we have included a red-lined version of the manuscript to track changes in the text.

Concluding, the manuscript presents a study of a well-known problem using an original but anticipated technique which is a slight modification of well-known results. Thus, I confirm that the paper is publishable in a scientific journal but does not meet the novelty criteria in this particular case.

We are thankful for recognizing the novelty of our idea. As the Reviewer says, the technique is original, and therefore new. It has not been demonstrated to date. The fact that it is a modification of existing techniques, and that it can be anticipated should make it look more appealing to broader audiences. As recognized by Reviewers #1 and #2, it could have high impacts on nonlinear optics study because of the large number of practical difficulties and fundamental limitations that hamper existing imaging methodologies. Therefore, along with Reviewers #1 and #2, we believe this work is suitable for publication in *Nature Communications*.

We hope that this satisfactorily addresses the concerns raised by the Reviewers and we thank them for their constructive comments.

Sincerely,

Lukasz Sterczewski

REVIEWERS' COMMENTS

Reviewer #1 (Remarks to the Author):

The authors addressed my concerns. I only have one minor follow-up:
On page 1: "However, high scan rates with EFXC are obtainable only with multi-GHz repetition-rate sources." It would be nice to cite [3,4] again in this sentence.

Reviewer #2 (Remarks to the Author):

The authors have satisfactorily addressed all my concerns in the revised manuscript. I would recommend its publication without any conservation.

Reviewer #3 (Remarks to the Author):

Despite the detailed reply presented by the authors, to my regret, I do not find the arguments convincing. I believe that the manuscript now is in the pre-publishing shape, however, the lack of sufficient novelty makes it suitable only for a more specialized journal.

Indeed, after some additional search, I found that this experimental scheme has been investigated > 20 years ago. To my point of view, the following list of 5 papers covers almost all the novelty aspects claimed by the authors:

This article discusses a similar experimental scheme in the context of optical sampling. Alone, it significantly obscures the novelty of the present manuscript.

[1] K. Kikuchi, Optical Sampling System at 1.5 [Micro Sign]m Using Two Photon Absorption in Si Avalanche Photodiode, *Electron. Lett.* 34, 1354 (1998).

This article investigates the phase-insensitivity of the exact scheme proposed by the authors:

[2] R. Salem and T. E. Murphy, Polarization-Insensitive Cross Correlation Using Two-Photon Absorption in a Silicon Photodiode, *Opt. Lett.* 29, 1524 (2004).

This one uses this scheme for the optical clock recovery:

[3] R. Salem and T. E. Murphy, Broad-Band Optical Clock Recovery System Using Two-Photon Absorption, *IEEE Photonics Technology Letters* 16, 2141 (2004).

Here the optical sampling at high rates is discussed:

[4] P. J. Maguire, L. P. Barry, T. Krug, J. O. Dowd, M. Lynch, A. L. Bradley, J. F. Donegan, and H. Folliot, Direct Measurement of a High-Speed (>100Gbit/s) OTDM Data Signal Utilising Two-Photon Absorption in a Semiconductor Microcavity, in *2005 IEEE LEOS Annual Meeting Conference Proceedings* (2005), pp. 142–143.

The reference below is a review article that mentions some of the relevant advances:

[5] A. Hayat, A. Nevet, P. Ginzburg, and M. Orenstein, Applications of Two-Photon Processes in Semiconductor Photonic Devices: Invited Review, *Semicond. Sci. Technol.* 26, 083001 (2011).

Finally, I have to admit that this experimental scheme has never been applied to the case of the CGLE soliton molecules, to the best of my knowledge. Thus, I confirm my conclusions made earlier: "the paper is publishable in a scientific journal but does not meet the novelty criteria in this particular case."

Dear Editor, dear Reviewers,

We would like to thank the Reviewers for providing insightful comments and suggestions that helped us strengthen the manuscript. Below, are the detailed changes in the manuscript addressing the Reviewers' comments (our responses are shown in *blue*).

Reviewer #1 (Remarks to the Author):

The authors addressed my concerns. I only have one minor follow-up:

On page 1: "However, high scan rates with EFXC are obtainable only with multi-GHz repetition-rate sources." It would be nice to cite [3,4] again in this sentence.

The suggested references have been included in the revised manuscript.

Reviewer #2 (Remarks to the Author):

The authors have satisfactorily addressed all my concerns in the revised manuscript. I would recommend its publication without any conservation.

We are thankful for the constructive comments and appreciation of our work. Encouraged by Reviewer's earlier questions, we have put considerable effort to improve the sensitivity to make the imaging technique suitable for unamplified oscillators or low-energy pulses.

Reviewer #3 (Remarks to the Author):

Despite the detailed reply presented by the authors, to my regret, I do not find the arguments convincing. I believe that the manuscript now is in the pre-publishing shape, however, the lack of sufficient novelty makes it suitable only for a more specialized journal.

Indeed, after some additional search, I found that this experimental scheme has been investigated > 20 years ago. To my point of view, the following list of 5 papers covers almost all the novelty aspects claimed by the authors:

This article discusses a similar experimental scheme in the context of optical sampling. Alone, it significantly obscures the novelty of the present manuscript.

[1] K. Kikuchi, Optical Sampling System at 1.5 [Micro Sign]m Using Two Photon Absorption in Si Avalanche Photodiode, Electron. Lett. 34, 1354 (1998).

This article investigates the phase-insensitivity of the exact scheme proposed by the authors:

[2] R. Salem and T. E. Murphy, Polarization-Insensitive Cross Correlation Using Two-Photon Absorption in a Silicon Photodiode, Opt. Lett. 29, 1524 (2004).

This one uses this scheme for the optical clock recovery:

[3] R. Salem and T. E. Murphy, Broad-Band Optical Clock Recovery System Using Two-Photon Absorption, IEEE Photonics Technology Letters 16, 2141 (2004).

Here the optical sampling at high rates is discussed:

[4] P. J. Maguire, L. P. Barry, T. Krug, J. O. Dowd, M. Lynch, A. L. Bradley, J. F. Donegan, and H. Folliot, Direct Measurement of a High-Speed (>100Gbit/s) OTDM Data Signal Utilising Two-Photon Absorption in a Semiconductor Microcavity, in 2005 IEEE LEOS Annual Meeting Conference Proceedings (2005), pp. 142–143.

The reference below is a review article that mentions some of the relevant advances:

[5] A. Hayat, A. Nevet, P. Ginzburg, and M. Orenstein, Applications of Two-Photon Processes in Semiconductor Photonic Devices: Invited Review, Semicond. Sci. Technol. 26, 083001 (2011).

Finally, I have to admit that this experimental scheme has never been applied to the case of the CGLE soliton molecules, to the best of my knowledge. Thus, I confirm my conclusions made earlier: “the paper is publishable in a scientific journal but does not meet the novelty criteria in this particular case.”

In the concluding sentences, the Reviewer admits that prior to our work, two-photon imaging of CGLE soliton molecules has never been performed. Starting from the abstract, we highlight that it is the *application* of the 2PA process to study this exciting ultrafast phenomenon that makes our work relevant and novel. Nowhere in the manuscript do we claim being first to use 2PA to perform cross-correlation measurements. We feel that comments related to the insufficient novelty of the experimental scheme are inappropriate. Even in the first-submitted manuscript, we put the following sentence in the introduction:

“(…) the imaging technique builds on the two-photon detection ranging concept by Wright et al.³⁵”

Also this sentence acknowledges prior advances:

“To bypass these limitations and unlock the kHz- to sub-MHz rate imaging potential, in this Article we adapt the non-interferometric intensity cross-correlation (IXC) technique³² to the problem of dynamic soliton imaging.”

The experimental scheme and prior work was therefore properly acknowledged from the beginning. Nevertheless, to satisfy the Reviewer’s requirement to add a historical context, some of the suggested references have been included in the revised manuscript. We find that the relevant ones are here Ref. 1, 2, and 4. Reference 3 discusses the use of the 2PA process to study temporal overlap between two nearly synchronized train pulses for clock recovery, while Reference 5 does not talk about asynchronous interaction between pulses lasers at all. Instead, it only discusses the implementation of an intensity autocorrelator and provides an example from Ref. 3. No reference, however, talks about the observation or characterization multi-pulse aggregates or soliton molecules. The modified sentence now reads:

“In this work, to bypass these limitations and unlock the kHz- to sub-MHz rate imaging potential, we adapt the non-interferometric intensity cross-correlation (IXC) technique³²⁻³⁵ to the problem of dynamic soliton imaging.” [new references have been included after the word “technique”].

Our work should be seen as a paradigm shift in the diagnostics of pulsed lasers utilizing optical fibers or microresonator cavities. Particularly the feasibility of leveraging telecom-wavelength lasers to probe

pulsed sources operating in the emerging 2-micron region (demonstrated here) even at highly dissimilar repetition rates (from a conventional dual-comb perspective) is highly attractive. Also an extension to the mid-IR region, urgently needed by the laser community, is quite straightforward via simple frequency translation of a 1.5 micron laser through soliton self-frequency shift merged with a suitable-bandgap detector (i.e. extended InGaAs).

Thanks to our work, it is likely that some of the early references suggested by the author will be re-discovered when put in a different context like laser pulse diagnostics. The *application* of the 2PA process to study soliton molecules, which the Reviewer agrees is unprecedented, fulfills the novelty criterion of the journal.

We hope that this satisfactorily addresses the concerns raised by the Reviewers and we thank them for their constructive comments.

Sincerely,

Lukasz Sterczewski